# Classification of chiral fermionic CFTs of central charge $\leq 16$

Philip Boyle Smith[1], Ying-Hsuan Lin[2], Yuji Tachikawa[1], and Yunqin Zheng[1,3]

1    Kavli Institute for the Physics and Mathematics of the Universe (WPI),
University of Tokyo, Kashiwa, Chiba 277-8583, Japan
2    Jefferson Physical Laboratory, Harvard University, Cambridge, MA 02138, USA
3    Institute for Solid State Physics, University of Tokyo, Kashiwa, Chiba 277-8583, Japan

We classify two-dimensional purely chiral unitary conformal field theories which are defined on two-dimensional surfaces equipped with spin structure and have central charge less than or equal to 16, and discuss their duality webs. This result can be used to confirm that the list of non-supersymmetric ten-dimensional heterotic string theories found in the late 1980s is complete and does not miss any exotic example.

# 1 Introduction and Summary

Two-dimensional unitary conformal field theories (CFTs) in general contain both left-moving (or chiral) and right-moving (or anti-chiral) degrees of freedom, with possibly distinct central charges $c_L$ and $c_R$.[1] They are known to describe universality classes of many critical systems, and it is a classic result that such theories with $c_L = c_R < 1$ can be completely classified [CIZ87, Kat87]. More precisely[2], what these references classified were *bosonic* CFTs, i.e. theories that are defined

---

[1]We assume the unitarity of theories throughout this paper. Non-unitary 2d CFTs also play important roles in the continuum limit of classical two-dimensional statistical physics, but they are outside of the concern of our paper.

[2]In this paper we use a recent understanding of quantum field theories (QFTs) in that to specify a single QFT we need to first specify the spacetime structure on which it depends. This spacetime structure can simply be an orientation, or it can include a spin structure. As another example, a parity-invariant theory in the traditional language can then be regarded as a QFT which does not require a specification of an orientation. Even after the specification of

and modular invariant on surfaces without spin structure. We can also consider *fermionic* CFTs, which are defined and modular invariant on surfaces equipped with spin structure, and those with $c_L = c_R < 1$ have been similarly classified at the physical level of rigor[3] [Pet88, RW20, HNT20, Kul20]. It is often stated that CFTs with $c_L = c_R = 1$ have been classified [Gin88, Kir89, Kir88], and CFTs with $c_L = c_R > 1$ are extremely rich and have defied any attempt at classification.

We can also consider theories with $c_L \neq c_R$. We require our CFTs to be modular invariant, in the sense that we have a partition function, rather than a partition vector, on a closed two-dimensional surface, and that the large diffeomorphisms act by phases and not by matrices. Among such CFTs, purely chiral CFTs, i.e. those with $c_L > 0$ and $c_R = 0$, have been used in theoretical physics in a different context. For example, purely chiral bosonic CFTs with $c_L = 16$ can be used as the worldsheet degrees of freedom for spacetime-supersymmetric heterotic string theories [GHMR85a]. Two such examples at $c_L = 16$ have long been known, one with $E_8 \times E_8$ symmetry and another with $\mathrm{Spin}(32)/\mathbb{Z}_2$ symmetry.[4]

In general, chiral operators in any CFT form a tight mathematical structure known as a chiral algebra in theoretical physics and formalized as a vertex operator algebra in mathematics. In a purely chiral CFT, the generating function of this algebra needs to be modular invariant by itself, which puts very strong constraints on such systems. This has allowed mathematicians to classify chiral bosonic CFTs with $c_L = 16$ in this century [DM02], without assuming that they are obtained from either a lattice or a free fermionic construction, thereby confirming that the chiral bosonic CFTs with $E_8 \times E_8$ and $\mathrm{Spin}(32)/\mathbb{Z}_2$ symmetries are the only possible ones.[5]

The aim of this paper is to perform a similar classification of chiral fermionic CFTs with $c_L \leq 16$ and $c_R = 0$, without assuming that they are obtained by a lattice or a free fermionic construction.[6] We find that any such CFT is a product of the basic ones listed in Table 1.[7]

---

the spacetime structure, the partition function of the theory can have a specific form of phase ambiguity dictated by its gravitational anomaly. A modular-invariant CFT in the traditional sense, whose partition function is invariant under the whole of $SL(2, \mathbb{Z})$, is a special case of a bosonic CFT with zero gravitational anomaly under the more modern understanding.

[3]This phrase will not be repeated further, as the discussions and derivations in this paper will be at this physics level of rigor.

[4] The Lie group $\mathrm{Spin}(4k)$ has center $\mathbb{Z}_2 \times \mathbb{Z}_2$, so we can consider three $\mathbb{Z}_2$ quotients. If the quotient by one of them gives $\mathrm{SO}(4k)$, then the quotient by either of the other two gives what we exclusively call $\mathrm{Spin}(4k)/\mathbb{Z}_2$.

[5]This can be considered as a worldsheet counterpart to string universality in ten dimensions, studied using supergravity in [ADT10].

[6]The construction of chiral fermionic CFTs from an odd self-dual lattice has a long history, see e.g. [LSW89, KY23]. For a recent take on free fermionic construction, also see e.g. [GJF18].

[7] The theory of a single Majorana-Weyl fermion $\psi$, or any other CFT whose $c_L - c_R$ is not an integer but a half-integer, has the following subtlety. Let $\chi_{\mathrm{NS}}^{\mathrm{even}}$ and $\chi_{\mathrm{NS}}^{\mathrm{odd}}$ be the character of the NS sector, restricted to the part $(-1)^F$ being even and odd, respectively. Then the $\mathcal{S}$ transformation of $\chi_{\mathrm{NS}}^{\mathrm{even}} - \chi_{\mathrm{NS}}^{\mathrm{odd}}$ corresponds to the partition function of the theory on a torus, where the spatial circle is in the R-sector and the spin structure around the temporal direction is anti-periodic. Its $q$-expansion coefficients are supposed to count the number of states in the R-sector with each $L_0$ eigenvalue, but they actually are $\sqrt{2}$ times non-negative integers.

This is essentially due to the fact that a Majorana-Weyl fermion on the R-sector circle gives rise to a single Majorana fermion zero mode $\psi_0$. If we have two such zero modes $\psi_0$ and $\psi'_0$, they can be realized on a two-dimensional Hilbert space with $\psi_0 := \sigma_1$, $\psi'_0 := \sigma_2$ and $(-1)^F := \sigma_3$. But a single $\psi_0$ would then require a Hilbert space of dimension $\sqrt{2}$, if we require a tensor-product decomposition into two factors with the same dimension. For a detailed pedagogical

As we will recall in more detail below, any chiral bosonic CFT with $c_L = 16$ leads to a 10-dimensional spacetime-supersymmetric heterotic string, whereas any chiral fermionic CFT with $c_L = 16$ leads to a 10-dimensional spacetime-non-supersymmetric heterotic string. Our classification then implies that the list of 10-dimensional spacetime-non-supersymmetric heterotic strings constructed in the mid-1980s [AGGMV86, DH86, SW86, KLT86] is actually complete.

The rest of the paper is organized as follows. In Sec. 2, we explain the strategy employed for the task at hand. We will see that the classification of chiral fermionic CFTs reduces to the classification of non-anomalous $\mathbb{Z}_2$ actions on chiral bosonic CFTs.[8] In Sec. 3, we will quickly review the classification of chiral bosonic CFTs with $c_L \leq 16$, carried out mathematically in [DM02], in a language more palatable to physicists. In Sec. 4 we will then classify all possible non-anomalous $\mathbb{Z}_2$ actions on them. In Sec. 5, we use the list of $\mathbb{Z}_2$ actions on chiral bosonic CFTs to construct the corresponding chiral fermionic CFTs, thus completing our classification program. We will conclude our paper in Sec. 6 by discussing the implications for heterotic string theories in ten dimensions or less.

We also have a few technical appendices. In App. A and B, we provide the decomposition of the even self-dual lattices of type $E_8$ and $D_{16}$ with respect to the $\mathbb{Z}_2$ actions we use in detail.

In some sense, all the results in this paper can already be found scattered throughout the literature. Therefore, the authors do not claim to have made a significant new discovery. Rather, the authors would like to gather this scattered information into a single place for future reference. For this purpose, the paper is written aiming for pedagogy and self-containedness.

**Conventions:**  In this paper, we mostly use traditional physicists' convention to ignore the difference between Lie groups sharing the same Lie algebra, unless necessary for context and clarity. This is due to the difficulty in actually pinning down the correct global structure of the group. For example, the theory often known as the $E_8 \times E_8$ theory has the symmetry group $(E_8 \times E_8) \rtimes \mathbb{Z}_2$. Also, with the modern terminology, a fermionic theory with NS sector states transforming under the group $G$ and with R sector states transforming under a nontrivial extension $G'$ of $G$ by $(-1)^F$, is said to have the symmetry group $G$ and the fact that the action on the R-sector states is extended is ascribed to the mixed anomaly between $G$ and the spacetime symmetry.

The only place we actually use the particular global forms is Sec. 4.2 where the $\mathbb{Z}_2$ actions on the bosonic $c = 16$ theories are classified. Even there it is only used indirectly, since we first consider order-2 elements in the automorphism group of Lie algebras, and then check whether they lift to order-2 operations acting on the representations contained in the CFT.

---

discussion of this subtlety, see e.g. [Wit23, Sec. 2.1] and [FHT24].

This subtlety in the R-sector of a theory with half-integer $c_L - c_R$ is now understood as a type of a gravitational anomaly of fermionic theories defined on two-dimensional spacetimes equipped with spin structure. In these cases, we take the convention that the $\mathcal{S}$ transformation of $\chi_{\mathrm{NS}}^{\mathrm{even}} - \chi_{\mathrm{NS}}^{\mathrm{odd}}$ is $\sqrt{2}\chi_{\mathrm{R}}$, where $\chi_{\mathrm{R}}$ has a $q$-expansion with non-negative integer coefficients; it is not possible to assign the fermionic parity $(-1)^F$ to the states counted by $\chi_{\mathrm{R}}$.

[8]A very early appearance of this construction can be found in [DGH88], where a fermionized version of the chiral Monster theory with $c = 24$ was found to have supersymmetry.

# Table 1: List of non-product chiral fermionic CFTs with $c \leq 16$.

| $c$ | name | $\chi_{\mathrm{NS}}^{\mathrm{even}}$ | $\chi_{\mathrm{NS}}^{\mathrm{odd}}$ | $\chi_{\mathrm{R}}^{\mathrm{even}}$ | $\chi_{\mathrm{R}}^{\mathrm{odd}}$ |
|---|---|---|---|---|---|
| $\frac{1}{2}$ | $\psi$ | $\chi_0$ | $\chi_{1/2}$ | $\chi_{1/16}$ | |
| $8$ | $(E_8)_1$ | $\chi_{\mathbf{1}}$ | $0$ | $\chi_{\mathbf{1}}$ | $0$ |
| $12$ | $\overline{(D_{12})_1}$ | $\chi_{\mathbf{1}}$ | $\chi_S$ | $\chi_{\mathbf{24}}$ | $\chi_C$ |
| $14$ | $\overline{(E_7)_1 \times (E_7)_1}$ | $\chi_{\mathbf{1}}\chi_{\mathbf{1}'}$ | $\chi_{\mathbf{56}}\chi_{\mathbf{56}'}$ | $\chi_{\mathbf{56}}\chi_{\mathbf{1}'}$ | $\chi_{\mathbf{1}}\chi_{\mathbf{56}'}$ |
| $15$ | $\overline{(A_{15})_1}$ | $\chi_{\mathbf{1}} + \chi_{\wedge^8\mathbf{16}}$ | $\chi_{\wedge^4\mathbf{16}} + \chi_{\wedge^{12}\mathbf{16}}$ | $\chi_{\wedge^2\mathbf{16}} + \chi_{\wedge^{10}\mathbf{16}}$ | $\chi_{\wedge^6\mathbf{16}} + \chi_{\wedge^{14}\mathbf{16}}$ |
| $\frac{31}{2}$ | $\overline{(E_8)_2}$ | $\chi_{\mathbf{1}}$ | $\chi_{\mathbf{3875}}$ | $\chi_{\mathbf{248}}$ | |
| $16$ | $\overline{(D_8)_1 \times (D_8)_1}$ | $\chi_{\mathbf{1}}\chi_{\mathbf{1}'} + \chi_S\chi_{S'}$ | $\chi_C\chi_{\mathbf{16}'} + \chi_{\mathbf{16}}\chi_{C'}$ | $\chi_{\mathbf{16}}\chi_{\mathbf{16}'} + \chi_C\chi_{C'}$ | $\chi_S\chi_{\mathbf{1}'} + \chi_{\mathbf{1}}\chi_{S'}$ |
| $16$ | $\overline{(D_{16})_1}$ | $\chi_{\mathbf{1}} + \chi_S$ | $0$ | $\chi_{\mathbf{1}} + \chi_S$ | $0$ |

EXPLANATION OF THE TABLE

- In the list, we first specify the central charge $c$ and the name we assign to each theory. The name $\psi$ is for the theory of a free Majorana Weyl fermion. The other names are based on the largest affine symmetry $G_k$ contained. When the theory is not simply the vacuum module of the affine symmetry $G_k$ but an *extension*, we use the notation $\overline{G_k}$ to emphasize this fact. The last theory $\overline{(D_{16})_1}$ is also denoted by $(\mathrm{Spin}(32)/\mathbb{Z}_2)_1$.

- Bosonic theories are shaded.

- $\chi_{\mathrm{NS,R}}^{\mathrm{even,odd}}$ are the characters of the Hilbert space on $S^1$ with NS/R spin structure, restricted to the eigenspace of $(-1)^F$ being even/odd.

- When $c$ is not an integer, $(-1)^F$ on the R-sector is not well defined, and therefore $\chi_{\mathrm{R}}^{\mathrm{even,odd}}$ are not listed separately. For more on this point, see the footnote 7.

- We do not separately list a theory and the theory obtained by multiplying by the Arf theory, for which $\chi_{\mathrm{R}}^{\mathrm{even}}$ and $\chi_{\mathrm{R}}^{\mathrm{odd}}$ are reversed.

- The characters $\chi_0$, $\chi_{1/2}$, $\chi_{1/16}$ are the Virasoro characters at $c = 1/2$.

- Other characters are affine characters, where the subscript specifies the irreducible representation in which the highest weight vector transforms under the finite-dimensional part of the symmetry. Primed representations are for the second factor in the affine symmetry.

- To specify an irreducible representation, we typically employ its dimension, except for the spinor $S$ and conjugate spinor $C$ representations of $D_n = \mathrm{SO}(2n)$, and for $\wedge^n\mathbf{16}$ for the $n$-th antisymmetric power of the fundamental representation of $A_{15} = \mathrm{SU}(16)$.

**Note:** The authors learned that mathematicians Gerald Höhn and Sven Möller have carried out the classification up to $c \leq 24$ rigorously using basically the same method [HM23], and that Brandon Rayhaun did the same classification up to $c \leq 23$ using a different method [Ray23]. G. Höhn and S. Möller also informed the authors that Höhn already had classified these theories mathematically up to $c \leq 31/2$ as Satz 3.2.4 of [Höh95] modulo some physics assumptions, and that the classification up to $c \leq 12$ was mathematically done in [CDR17]. The authors thank G. Höhn, S. Möller, and B. Rayhaun for discussions and also for coordinating the submission to the arXiv on the same date. They also thank Y. Moriwaki for notifying them about [HM23] in the first place.

# 2    Strategy

Our aim is to classify chiral fermionic CFTs without assuming that they admit a lattice and/or a free-fermionic construction. This is made possible by recent developments in our understanding of anomalies and of fermionization.

## 2.1    Mapping fermionic theories to bosonic theories

We start by noting that a general CFT with left- and right-moving central charges given by $c_L$ and $c_R$ has a gravitational anomaly specified by its anomaly polynomial $(c_L - c_R)p_1/24$. By the general theory of anomalies of unitary theories [Fre14, FH16], the gravitational anomaly of a unitary[9] bosonic (or fermionic) two-dimensional theory is such that the integral of the anomaly polynomial on an arbitrary oriented (or spin) four-dimensional manifold $M$ is integral.[10] Now, the integral of the Pontrjagin class $p_1$ on $M$ is three times its signature, and the signature of an oriented 4-manifold can be any integer, while the signature of a spin manifold is a multiple of 16. Therefore, the anomaly polynomial of a bosonic theory is an integer multiple of $p_1/3$, while that of a fermionic theory is an integer multiple of $p_1/48$. This means that $c_L - c_R$ is an integer multiple of $8$ in any bosonic CFT, while it is an integer multiple of $1/2$ in any fermionic CFT.

The theory of a single Majorana-Weyl fermion is a fermionic CFT with $(c_L, c_R) = (1/2, 0)$. Let us denote it simply by $\psi$. More generally, we can consider a $(c_L, c_R) = (k/2, 0)$ theory of $k$ free Majorana-Weyl fermions, which we denote by $k\psi$.[11] Given any fermionic CFT $T$ with

---

[9]A non-unitary theory can have anomalies not covered in the framework of [FH16]. See [CL20] and [HTY20, App. E]. This is one point in our argument where the unitary assumption is crucial.

[10]This condition might not be apparent from the discussion in [FH16], but is implicitly contained in it in the following manner. The discussion of [FH16] identifies the anomaly of a $n$-dimensional theory of spacetime structure $S$ to be given by an invertible $(n+1)$-dimensional theory, which is classified by the Anderson dual of the $S$-bordism groups. The anomaly polynomial is given by the non-torsion part of the Anderson dual, and as explained for physicists e.g. in [LOT20, Sec. 2.8], it has the requirement that it integrates to an integer on any $(n+2)$-manifold with $S$ structure.

[11]In our paper, we consider the combination of two theories $A$ and $B$ as taking products, since the Hilbert space of the combined system is the tensor product of the Hilbert spaces of individual theories. From this viewpoint the notation $\psi^k$ or $\psi^{\otimes k}$ might be more logical. We use the notation $k\psi$ as this seems more customary in the literature.

arbitrary $c_L - c_R = n/2$, we can then multiply $T$ by a number of copies of $\psi$ so that $c_L - c_R$ for the product theory $T \times k\psi$ is a multiple of 8. We denote this combination $T \times k\psi$ by $T_F$.

We can then invoke the modern understanding of bosonization and fermionization of two-dimensional quantum field theories [Tac18, KTT19], which says that the summation over the spin structure of a fermionic theory $T_F$ whose $c_L - c_R$ is a multiple of eight gives a bosonic theory $T_B$ with a non-anomalous $\mathbb{Z}_2$ symmetry $g$:[12]

$$T_B = T_F/(-1)^F. \tag{2.1}$$

Conversely, the fermionic theory $T_F$ can be reconstructed from the bosonic theory $T_B$ together with the $\mathbb{Z}_2$ action $g$ by orbifolding in the following way:

$$T_F = [T_B \times Q]/\mathbb{Z}_2. \tag{2.2}$$

Here, $Q$ is a spin invertible theory with $\mathbb{Z}_2$ symmetry whose partition function on a surface with spin structure $q$ and the $\mathbb{Z}_2$ symmetry background $a$ is $(-1)^{q(a)} := (-1)^{\mathrm{Arf}(q+a)-\mathrm{Arf}(q)}$. Here $q(a)$ is the quadratic refinement of the intersection pairing associated to the spin structure $q$, and Arf is the Arf invariant. We then perform the orbifolding with respect to the diagonal combination of the $\mathbb{Z}_2$ action $g$ on $T_B$ and the $\mathbb{Z}_2$ action on the theory $Q$.

The discussion so far means that any fermionic CFT $T$ can be put in the following form:

$$T \times k\psi = [T_B \times Q]/\mathbb{Z}_2 \tag{2.3}$$

for some bosonic theory $T_B$ with a non-anomalous $\mathbb{Z}_2$ symmetry $g$, where $k$ is the smallest (or indeed any) non-negative integer such that $k/2 + c_L(T) - c_R(T)$ is a multiple of 8.

Note that we have not imposed the condition that our original theory $T$ is chiral. When $T$ is chiral, we easily see that $T_B$ is also chiral. Therefore, to classify chiral fermionic CFTs with $c_L \leq 8n$, we only have to classify chiral bosonic CFTs with $c_L \leq 8n$ and non-anomalous $\mathbb{Z}_2$ actions on them.

## 2.2 Dimension-0 operators in fermionic theories

Before proceeding, we need to deal with the subtlety that bosonization/fermionization does not always preserve the property that the theory contains a unique vacuum state. For example, when $T_F$ is the trivial theory, $T_B$ has two vacua on $S^1$, one which is even under $g$ and another which is odd under $g$.[13]

---

[12]To see that this is possible, note that the partition function of a theory with $c_L - c_R = n/2$ is a vector in the one-dimensional Hilbert space of the bulk spin invertible theory $\mathrm{SO}(n)_1$. When $n$ is a multiple of 16, the bulk theory is actually bosonic and does not require the spin structure. Therefore, the partition function of $T_F$ on a 2d surface $S$ with a spin structure $q$ takes values in a single one-dimensional Hilbert space independent of $q$, when $n$ is a multiple of 16. This allows us to sum over the spin structure without problems, resulting in a bosonic 2d theory. The summation over spin structure is possible when $n$ is 8 modulo 16, but the result is still a fermionic theory. The details will be discussed in a forthcoming paper [BSZ24].

[13] A trivial fermionic theory $T_F$ has a single $(-1)^F$-even state in the NS sector, together with a $(-1)^F$-even state in the R sector; the latter is necessary to satisfy modular invariance. Under the quotient (2.1), the Hilbert space of

To study these issues, let us first analyze the dimension-0 operators in general fermionic theories. Suppose we are given a fermionic CFT with a unique dimension-0 operator in the NS sector, the identity, together with $n_R$ dimension-0 operators in the R sector. The possible values of $n_R$ are 0 and 1. To see this, let us consider the torus partition functions $\mathcal{Z}_{NS/R}^{NS/R}$ with different spin structures, where the superscript and subscript represent the fermion boundary conditions (NS for antiperiodic and R for periodic) in time and space, respectively. Under a modular transformation,

$$\mathcal{Z}_{NS}^{NS}(\tau) - \mathcal{Z}_{NS}^{R}(\tau) = \mathcal{Z}_{NS}^{NS}(-\tfrac{1}{\tau}) - \mathcal{Z}_{R}^{NS}(-\tfrac{1}{\tau}), \tag{2.4}$$

which in the Cardy regime $-\tfrac{1}{\tau} \sim +i\infty$ becomes

$$\mathrm{tr}_{NS}\big((1 - (-1)^F)q^{L_0 - c/24}\big) \sim (1 - n_R)e^{2\pi i c/24\tau} + \cdots. \tag{2.5}$$

If $n_R > 1$, then an inverse Laplace transform reveals a negative asymptotic density of NS sector states that are odd under fermion parity $(-1)^F$, contradicting unitarity.

If there are $n_{NS}$ dimension-0 operators in the NS sector, then the argument above gives $n_R \leq n_{NS}$. However, we can say more. The $n_{NS}$ dimension-0 operators form a commutative[14] Frobenius algebra under the operator product, and it is well-known that there always exists an idempotent basis $\{1_i \mid i = 1, \ldots, n_{NS}\}$ such that

$$1_i 1_j = 1_i \delta_{ij}. \tag{2.6}$$

Since both the NS and R Hilbert spaces are modules of this Frobenius algebra, the full theory can be decomposed into $n_{NS}$ 'universes' [TU19] (exact superselection sectors separated by domain walls of infinite tension), which we label by $i = 1, \ldots, n_{NS}$. Any correlator on a connected spacetime must vanish if it involves local operators from two or more universes. The full theory is said to be a direct sum of the component universes, where each universe has $n_{NS}^i = 1$ (counting the idempotent $1_i$ which acts as the identity operator within its universe) and $n_R^i = 0, 1$.

Let us now come back to the relation between $T_B$ and $T_F$, assuming $n_{NS} = 1$. It is straightforward to see that $T_B$ has a unique vacuum when $T_F$ does not have a dimension-0 operator in the R-sector.

Conversely, suppose $T_F$ has a dimension-0 operator in the R-sector. We can now use this operator to map any operator in the R-sector to an operator in the NS-sector and vice versa, establishing a 1-to-1 map between the R-sector states and the NS sector states, commuting with the Virasoro action. As the R-sector states have all integer spins, the NS sector states also have integer spins. This means that the restriction of the theory to the NS sector operators actually defines a bosonic theory, which we denote by $T_B'$. Depending on the $(-1)^F$ eigenvalue of the R-sector dimension-0 operator, $T_F$ is then the product of $T_B'$ with a completely trivial fermionic theory (whose partition function is always 1) or with the Arf theory (whose partition function is the Arf invariant). Barring these two degenerate cases, we are guaranteed that $T_B$ has a unique vacuum, and the $\mathbb{Z}_2$ symmetry $g$ acts nontrivially on $T_B$.

---

the bosonic theory $T_B$ inherits both states, resulting in a theory with two states of zero energy. In general, gauging a trivially-acting global symmetry group can result in a nontrivial theory. For example, 4d pure $SU(N)$ gauge theory is obtained by gauging a global $SU(N)$ symmetry acting on a completely trivial theory, and this is clearly highly nontrivial.

[14]They commute due to the spin-statistics theorem, as the dimension-0 operators are spinless. Here we use the unitarity condition in an essential way.

## 2.3 Applications to theories with low $c$

This program can be pursued below the central charge $c_L$ for which the list of all chiral bosonic CFTs is known. Presently, this restricts us to $c_L = 8, 16$ for which the classification can be found in [DM02], and $c_L = 24$ for which the list had originally been conjectured by Schellekens [Sch92] and later made into a mathematical theorem in [vEMS15, MS19, vELMS20, HM20]. As the list for $c_L = 24$ is considerably larger, and as the case $c_L \leq 16$ has a more immediate application to heterotic string theories, we restrict our attention to $c_L \leq 16$ in this paper.

# 3 Chiral bosonic CFTs with $c \leq 16$

In [DM02], it was shown that the only chiral bosonic CFT at $c = 8$ is the $E_8$ level 1 theory (also known as the $E_8$ lattice theory) and that the only two such CFTs at $c = 16$ are the properly extended $D_{16}$ level 1 theory (also known as the $D_{16}$ lattice theory) and the product of two copies of the $E_8$ level 1 theory. This result is proven without assuming that the CFTs are constructed via lattice or free fermionic constructions. Let us briefly recall their argument in our language.

## 3.1 General constraints on partition functions

We start with a general argument. We have seen that a chiral bosonic CFT has $c = 8n$ for an integer $n$. Let us then consider its partition function

$$\mathcal{Z}(\tau) = \operatorname{tr} q^{L_0 - c/24} = q^{-c/24}(1 + \mathcal{O}(q)). \tag{3.1}$$

In the case of the well-known bosonic CFT of $c = 8$ corresponding to the $E_8$ level 1 current algebra, its partition function $\mathcal{Z}_{E_8}(\tau)$ behaves under modular transformations as

$$\mathcal{Z}_{E_8}(\tau + 1) = e^{-2\pi i/3} \mathcal{Z}_{E_8}(\tau), \qquad \mathcal{Z}_{E_8}(-\tfrac{1}{\tau}) = \mathcal{Z}_{E_8}(\tau). \tag{3.2}$$

The phases in these relations come from a gravitational anomaly and are determined solely by the central charge. Therefore, we know that in general the partition function (3.1) behaves as

$$\mathcal{Z}(\tau + 1) = e^{-2\pi i n/3} \mathcal{Z}(\tau), \qquad \mathcal{Z}(-\tfrac{1}{\tau}) = \mathcal{Z}(\tau), \tag{3.3}$$

where $c = 8n$.

We now consider the function

$$W(\tau) := \eta(\tau)^{-16n} \mathcal{Z}(\tau) = q^{-n}(1 + \mathcal{O}(q)) \tag{3.4}$$

where

$$\eta(\tau) = q^{1/24} \prod_{n=1}^{\infty} (1 - q^n) \tag{3.5}$$

is the Dedekind eta function. Its modular transformations are given by

$$W(\tau + 1) = W(\tau), \qquad W(-\tfrac{1}{\tau}) = \tau^{-8n} W(\tau), \tag{3.6}$$

which means that it is a weakly-holomorphic modular form of weight $-8n$.

## 3.2 Applications to the case $c = 8$ or $16$

Any weakly-holomorphic modular form of weight $w$ is known[15] to be a $\mathbb{C}$-linear combination of monomials

$$E_4(\tau)^i E_6(\tau)^j \Delta(\tau)^k = q^k(1 + \mathcal{O}(q)) \tag{3.7}$$

where $i \geq 0$, $j \in \{0, 1\}$, $k \in \mathbb{Z}$ and $4i + 6j + 12k = w$. Here

$$E_4(\tau) = 1 + 240q + \cdots, \qquad E_6(\tau) = 1 - 504q + \cdots, \tag{3.8}$$

are the normalized Eisenstein series of weight $4$ and $6$, and

$$\Delta(\tau) = \frac{E_4(\tau)^3 - E_6(\tau)^2}{1728} = \eta(\tau)^{24} = q(1 - 24q + \cdots) \tag{3.9}$$

is the modular discriminant.

The information above imposes a very strong constraint on $W(\tau)$ when $c = 8$ or $c = 16$. Indeed, the leading pole given in (3.4) and the explicit basis (3.7) uniquely fix $W(\tau)$ to be

$$W(\tau) = \begin{cases} E_4(\tau)/\Delta(\tau) & (c = 8), \\ (E_4(\tau)/\Delta(\tau))^2 & (c = 16), \end{cases} \tag{3.10}$$

meaning that

$$\mathcal{Z}(\tau) = \begin{cases} E_4(\tau)/\eta(\tau)^8 = q^{-1/3}(1 + 248q + \cdots) & (c = 8), \\ E_4(\tau)^2/\eta(\tau)^{16} = q^{-2/3}(1 + 496q + \cdots) & (c = 16). \end{cases} \tag{3.11}$$

The analysis so far means that any chiral bosonic CFT at $c = 8$ has $248$ spin-1 currents, which necessarily form a reductive Lie algebra[16] $G$ with dimension $248$. We also know that the rank of $G$ is at most $8$, since any Lie algebra of rank $r$ contains a $U(1)^r$ subalgebra, which leads to $c \geq r$.[17] From this one easily concludes that the only possibility is to take $G = E_8$; see Sec. 3.3 below. The level of $E_8$ is fixed to be $1$ from the central charge. As the character of the affine algebra $E_8$ at level $1$ is $E_4(\tau)/\eta(\tau)^8$, we conclude that a chiral bosonic CFT at $c = 8$ is necessarily the $E_8$ level 1 theory.

For chiral bosonic theories at $c = 16$, one carries out the same analysis for $496$ currents with rank $16$; for details, see Sec. 3.3. The only possibilities are $G = E_8 \times E_8$ both at level 1 and $G = SO(32)$ again at level 1. For details, we refer the reader again to Sec. 3.3. The former is simply the product of two copies of the $E_8$ level 1 theory we just found.

---

[15]This follows from the standard fact that the ring of modular forms over $\mathbb{C}$ is a polynomial ring over $E_4(\tau)$ and $E_6(\tau)$, that the ring of weakly-holomorphic modular forms is given by inverting $\Delta(\tau)$, and the relation $1728\Delta(\tau) = E_4(\tau)^3 - E_6(\tau)^2$ to rewrite $E_6(\tau)^{n \geq 2}$ in terms of $E_4(\tau)$ and $\Delta(\tau)$. For a mathematical reference, the lecture notes by Zagier [Zag08] can be highly recommended.

[16]Here a reductive Lie algebra is a direct sum of simple and $U(1)$ components. This condition is equivalent to the existence of a positive-definite invariant inner product.

[17]The same technique of constraining 2d CFTs from the number of spin-1 states and the central charge was used in [BL20, DK22].

To analyze the latter, we note that the vacuum character $\chi_{\mathbf{1}}$ is not equivalent to the desired partition function $\mathcal{Z}(\tau) = E_4(\tau)^2/\eta(\tau)^{16}$, but we do have the equality[18]

$$\mathcal{Z}(\tau) = \chi_{\mathbf{1}}^{D_{16}} + \chi_S^{D_{16}} , \tag{3.12}$$

where $\chi_S^{D_{16}}$ is the level-1 character of $\mathrm{SO}(32)$ whose highest weight state is one of the spinor representations of $\mathrm{SO}(32)$. This uniquely fixes the decomposition of the Hilbert space of this theory under the $\mathrm{SO}(32)$ level-1 affine symmetry. To fix the full theory, we also need to fix the three-point functions. There are four types, $\langle\mathbf{111}\rangle$, $\langle\mathbf{11S}\rangle$, $\langle\mathbf{1SS}\rangle$, and $\langle\mathbf{SSS}\rangle$. The only ones allowed by the $\mathrm{SO}(32)$ symmetry are $\langle\mathbf{111}\rangle$ and $\langle\mathbf{1SS}\rangle$, each of which has a unique $\mathrm{SO}(32)$-invariant combination, and their normalizations are fixed by the two-point functions. Therefore there can be at most one such theory. The existence is guaranteed by constructing the theory via the even self-dual lattice of type $D_{16}$. This concludes the classification of chiral bosonic CFTs with $c \leq 16$. In the following, we denote the two theories of $c = 16$ by $(E_8)_1 \times (E_8)_1$ and $(\mathrm{Spin}(32)/\mathbb{Z}_2)_1$, respectively.

## 3.3 A group theory lemma

Here we give a proof of a statement used above, that the only reductive Lie algebra of rank at most 8 with dimension 248 is $E_8$, and similarly that the only reductive Lie algebras of rank at most 16 with dimension 496 are $E_8 \times E_8$ and $D_{16}$.

For this, we consider the quantity $\dim(G)/\operatorname{rank}(G)$ for a Lie algebra $G$. When $G$ is simple, it is known that [Ste59]

$$\dim(G)/\operatorname{rank}(G) = h(G) + 1 \tag{3.13}$$

where $h(G)$ is the (non-dual) Coxeter number. We also know that clearly

$$\dim(\mathrm{U}(1))/\operatorname{rank}(\mathrm{U}(1)) = 1. \tag{3.14}$$

For a direct sum of these simple or $\mathrm{U}(1)$ components, $\dim/\operatorname{rank}$ is a weighted average of $\dim/\operatorname{rank}$ of individual components. So, in order to have $\dim/\operatorname{rank} \geq 248/8 = 496/16 = 31$, we need to use at least one component whose $\dim/\operatorname{rank} \geq 31$. As $\mathrm{U}(1)$ does not satisfy this condition, we need to use at least one simple factor, and the only simple factors for which $\dim/\operatorname{rank} \geq 31$ and $\operatorname{rank} \leq 16$ are $E_8$, $B_{15}$, $C_{15}$, $D_{16}$ (for which $\dim/\operatorname{rank} = 31$) and $B_{16}$, $C_{16}$ (for which $\dim/\operatorname{rank} = 33$). Therefore, the only possibility at rank 8 is $E_8$, and the only possibilities at rank 16 are $E_8 \times E_8$ and $D_{16}$, as $B_{15} \times \mathrm{U}(1)$ and $C_{15} \times \mathrm{U}(1)$ do not have $\dim = 496$.

---

[18]This partition function is that of the $(\mathrm{Spin}(32)/\mathbb{Z}_2)_1$ chiral WZW model. By the bulk-boundary correspondence, such a theory lives on the boundary of a $(\mathrm{Spin}(32)/\mathbb{Z}_2)_1$ Chern-Simons theory. Note that the $\mathrm{Spin}(32)_1$ Chern-Simons theory is a non-invertible TQFT with four topological lines, labeled by the adjoint ($\mathbf{1}$), spinor ($S$), vector ($V$) and conjugate-spinor ($C$) representations of $\mathrm{Spin}(32)$, respectively. For the chiral WZW model to be modular invariant under $\mathcal{S}$ (rather than modular covariant), the bulk TQFT should be invertible. Thus one needs to condense one of the lines in $\mathrm{Spin}(32)_1$. Condensing the $V$ line results in an invertible spin-TQFT $(\mathrm{Spin}(32)/\mathbb{Z}_2^V)_1 = \mathrm{SO}(32)_1$ since $V$ has spin $\frac{1}{2}$. To get an invertible non-spin TQFT, one needs to condense the line in $S$, yielding $(\mathrm{Spin}(32)/\mathbb{Z}_2^S)_1$. Condensing the $C$ line is related by an outer automorphism and gives an equivalent theory.

# 4  $\mathbb{Z}_2$ actions on chiral bosonic CFTs with $c \leq 16$

In this section we classify all $\mathbb{Z}_2$ symmetries of the $(E_8)_1$, $(E_8)_1 \times (E_8)_1$ and $(\mathrm{Spin}(32)/\mathbb{Z}_2)_1$ theories. Our approach follows that of [BKN21], where the classification for $(E_8)_1$ was already carried out.

To start, we note that any symmetry of a CFT should act as a symmetry of the three-point functions of spin-1 currents, which form a Lie algebra $G$. Therefore, it acts as an automorphism of $G$. As the symmetry acting on the CFT is of order 2, the automorphism of $G$ is either of order 1 or of order 2. Therefore, what we are going to do is to classify such automorphisms of $G$, and then to study which of them lift to order-2 symmetries of the CFT.

Finite-order automorphisms of $G$ are fully classified by Kac's theorem [Kac94, §8.6].[19] We first review the essential ingredients of Kac's theorem and then apply them to classify the $\mathbb{Z}_2$ symmetries of the $(E_8)_1$, $(E_8)_1 \times (E_8)_1$ and $(\mathrm{Spin}(32)/\mathbb{Z}_2)_1$ theories. For each symmetry, we additionally determine whether or not it has an anomaly.

## 4.1  Kac's theorem

Kac's theorem [Kac94, §8.6] is a procedure for determining all automorphisms of a simple Lie algebra $G$ of a given, finite order $m \geq 1$. Although the most general version of Kac's theorem classifies both inner and outer automorphisms, we will only need to classify inner automorphisms, the reason for which will become clear later. We thus give a brief review of Kac's theorem for this special case.

**Ingredients:**  A simple Lie algebra $G$, and the desired order $m \geq 1$.

**Recipe:**  Let $\ell$ be the rank of $G$, and let $a_i$ ($i = 0, 1, ..., \ell$) be the marks on the affine Dynkin diagram of $\hat{G}$, i.e. the unique eigenvector with all positive entries of the affine Cartan matrix, normalized to have $a_0 = 1$. One begins by listing all solutions to the equation

$$m = \sum_{i=0}^{\ell} a_i s_i \tag{4.1}$$

where the $s_i$ are non-negative and relatively prime integers. For each solution, there is a standard order-$m$ inner automorphism of the Lie algebra given by

$$E_\alpha \to e^{2\pi i(x,\alpha)} E_\alpha \quad \text{where} \quad x = \frac{1}{m} \sum_{i=1}^{\ell} s_i \omega_i, \tag{4.2}$$

---

[19]The analysis of order-two automorphisms of $E_8$ and $\mathrm{Spin}(32)/\mathbb{Z}_2$ was done more directly in [BLP+96, Sec. 5.1]. The analysis for general $\mathrm{Spin}(4n)/\mathbb{Z}_2$ was also essentially performed in [McI99a, McI99b]. We use Kac's theorem for its uniformity.

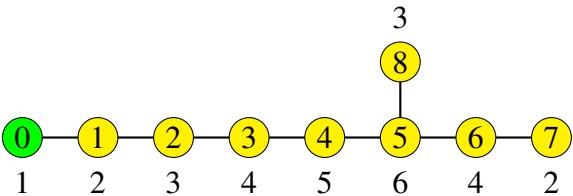

Figure 1: Affine Dynkin diagram of $E_8$. The inner number is the index $i$ of the node, while the outer number is the mark $a_i$.

where $\alpha$ is a root, and $\omega_i$ are the fundamental weights. Kac's theorem states that any order-$m$ inner automorphism is conjugate in $\mathrm{Aut}(G)$ to a standard one. Furthermore, two standard automorphisms are conjugate in $\mathrm{Aut}(G)$ if and only if the corresponding $s_i$ are related by a symmetry of the affine Dynkin diagram.

The invariant (fixed-point) Lie subalgebra also has a simple description in terms of the $s_i$. If there are $p$ nonvanishing $s_i$, then it is isomorphic to a direct sum of a $(p-1)$-dimensional center and a semisimple Lie algebra whose Dynkin diagram corresponds to the nodes with vanishing $s_i$.

When the automorphism is lifted to a $\mathbb{Z}_m$ symmetry of the CFT, it acts in the same way

$$\Gamma_\alpha \to e^{2\pi i(x,\alpha)}\Gamma_\alpha \tag{4.3}$$

on the vertex operators $\Gamma_\alpha$ with $\alpha$ in the root lattice. Note however that there may be further obstructions and choices involved in defining how the symmetry acts on the other vertex operators.

The anomaly of the $\mathbb{Z}_m$ symmetry is determined by the spin of the symmetry line [CLS+18] and in our case is given by the formula

$$h = \frac{(x,x)}{2} = \frac{1}{2m^2}\sum_{i,j=1}^{\ell} s_i s_j (\omega_i, \omega_j) \overset{ADE}{=} \frac{1}{2m^2}\sum_{i,j=1}^{\ell} s_i s_j (A^G)_{i,j}^{-1}, \tag{4.4}$$

where $A^G$ is the Cartan matrix of the Lie algebra $G$, and the last equality holds for simply-laced $G$, i.e. of type $A, D, E$. The $\mathbb{Z}_m$ symmetry is anomaly-free if $h = 0 \mod \frac{1}{m}$.

## 4.2 Classification of $\mathbb{Z}_2$ symmetries

### 4.2.1 $\mathbb{Z}_2$ symmetries in $(E_8)_1$ chiral CFT

The $E_8$ affine Dynkin diagram and the marks $a_i$ associated to the nodes are as shown in Fig. 1. Demanding $m = 2$ in (4.1), there are only two non-negative and relatively prime integer solutions for $s_i$,

$$\mathbb{Z}_2^a: \quad s_i = \begin{cases} 1, & i = 7, \\ 0, & \text{otherwise}, \end{cases} \qquad \mathbb{Z}_2^b: \quad s_i = \begin{cases} 1, & i = 1, \\ 0, & \text{otherwise}. \end{cases} \tag{4.5}$$

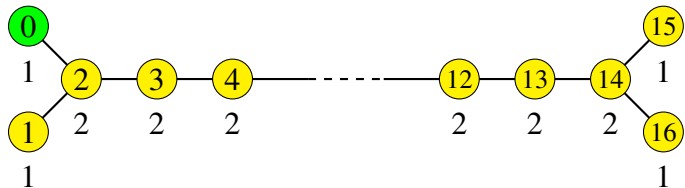

Figure 2: Affine Dynkin diagram of $D_{16} = \mathrm{SO}(32)$. The inner number is the index $i$ of the node, while the outer number is the mark $a_i$.

Each automorphism lifts to a unique symmetry of the CFT because $(E_8)_1$ only contains the identity module. From (4.4), the spins of the $\mathbb{Z}_2$ lines are

$$\mathbb{Z}_2^a: \quad h_a = \frac{(A^{E_8})_{77}^{-1}}{8} = 0 \bmod \frac{1}{2}, \qquad \mathbb{Z}_2^b: \quad h_b = \frac{(A^{E_8})_{11}^{-1}}{8} = \frac{1}{4} \bmod \frac{1}{2}, \qquad (4.6)$$

which implies that $\mathbb{Z}_2^a$ is anomaly-free and $\mathbb{Z}_2^b$ is anomalous. The invariant subalgebras are $D_8$ and $E_7 \times A_1$, respectively. One could then fermionize the $(E_8)_1$ chiral CFT using $\mathbb{Z}_2^a$. After gauging, the theory would have $(D_8)_1$ affine symmetry.

### 4.2.2 $\mathbb{Z}_2$ symmetries of $(E_8)_1 \times (E_8)_1$ chiral CFT

We can easily reuse our results from the previous section to find the order-two automorphisms of $E_8 \times E_8$. Let $g_i$ denote the element of $\mathrm{Aut}(E_8 \times E_8)$ given by acting with the automorphism $g \in \mathrm{Aut}(E_8)$ on the $i$th copy of $E_8$, $i = 1, 2$, and let $\sigma$ denote the exchange automorphism. Then the order-two automorphisms in $\mathrm{Aut}(E_8 \times E_8)$ are either of the form $g_1 h_2$, satisfying $g^2 = h^2 = 1$, or of the form $\sigma g_1 h_2$, satisfying $gh = 1$. Noting that the automorphism of the second form is conjugate to $\sigma$ alone, i.e. $\sigma g_1 h_2 = g_2 \sigma g_2^{-1}$, and also that $a_2$ is conjugate to $a_1$ by $\sigma$, i.e. $a_2 = \sigma a_1 \sigma^{-1}$, there are in total four non-anomalous conjugacy classes in $\mathrm{Aut}(E_8 \times E_8)$,

$$a_1, \quad a_1 a_2, \quad b_1 b_2, \quad \sigma. \qquad (4.7)$$

The first three belong to $\mathrm{Aut}(E_8) \times \mathrm{Aut}(E_8) \subset \mathrm{Aut}(E_8 \times E_8)$, and hence their anomalies follow from those of $\mathrm{Aut}(E_8)$ acting on a single copy of $(E_8)_1$. To check whether the $\mathbb{Z}_2$ generated by $\sigma$ is anomalous, we compute the torus partition function with a $\sigma$ line extended along the time cycle, $\mathcal{Z}_\sigma = E_4(\frac{\tau}{2})/\eta(\frac{\tau}{2})^8 = q^{-\frac{2}{3}}(q^{\frac{1}{2}} + 248q + \cdots)$; this computation is reviewed later in Sec. 5.2.4. This means that the spin of the state in the $\sigma$-twisted sector is half-integral, i.e.

$$h_\sigma = 0 \bmod \frac{1}{2}. \qquad (4.8)$$

Thus $\mathbb{Z}_2^\sigma$ is anomaly-free. One could then fermionize the $(E_8)_1 \times (E_8)_1$ chiral CFT by using each of $\mathbb{Z}_2^{a_1}$, $\mathbb{Z}_2^{a_1 a_2}$, $\mathbb{Z}_2^{b_1 b_2}$, and $\mathbb{Z}_2^\sigma$, whose invariant affine symmetries are $D_8 \times E_8'$, $D_8 \times D_8'$, $A_1 \times E_7 \times A_1' \times E_7'$, and $(E_8)_2$, respectively. We have used the prime to denote the second factor.

### 4.2.3 $\mathbb{Z}_2$ symmetries of $(\mathrm{Spin}(32)/\mathbb{Z}_2)_1$ chiral CFT

Here, a new subtlety arises, which is that $\mathrm{Aut}(D_{16})$ consists of both inner and outer automorphisms, but only $\mathrm{Inn}(D_{16})$ corresponds to actual symmetries of the CFT. This is because the $(\mathrm{Spin}(32)/\mathbb{Z}_2)_1$ theory contains the trivial and spinor module, and the spinor representation is not invariant under an outer automorphism.

Our immediate task is therefore to classify the order-two elements of $\mathrm{Inn}(D_{16})$ up to conjugacy. For the affine Dynkin diagram of $D_{16}$, the marks are as shown in Fig. 2. By solving (4.1) with $m = 2$, there are $13 + 6 = 19$ standard order-two inner automorphisms of $D_{16}$, specified by the following choices of $s_i$:

$$\langle n \rangle : \quad s_i = \begin{cases} 1, & i = n, \\ 0, & \text{otherwise,} \end{cases} \qquad n \in \{2, 3, ..., 14\},$$

$$\langle n_1, n_2 \rangle : \quad s_i = \begin{cases} 1, & i = n_1, n_2, \\ 0, & \text{otherwise,} \end{cases} \qquad n_1, n_2 \in \{0, 1, 15, 16\}; \quad n_1 < n_2. \tag{4.9}$$

Up to Dynkin diagram symmetries, there are $7 + 2 = 9$ equivalence classes, namely $n = 2, \ldots, 8$ and $(n_1, n_2) = (0, 1), (0, 16)$. But this measures conjugacy in $\mathrm{Aut}(D_{16})$, not $\mathrm{Inn}(D_{16})$. To determine the latter, we write each automorphism explicitly as a conjugation by an element $X \in \mathrm{SO}(32)$:

$$
\begin{aligned}
X_{\langle n \rangle} &= \mathrm{diag}(+^{2n}, -^{32-2n}), & X_{\langle 0,16 \rangle} &= \epsilon^{\oplus 16}, \\
X_{\langle 0,1 \rangle} &= \mathrm{diag}(+^2, -^{30}), & X_{\langle 0,15 \rangle} &= \epsilon^{\oplus 15} \oplus (-\epsilon), \\
X_{\langle 15,16 \rangle} &= \mathrm{diag}(+^{30}, -^2), & X_{\langle 1,16 \rangle} &= (-\epsilon) \oplus \epsilon^{\oplus 15}, \\
& & X_{\langle 1,15 \rangle} &= (-\epsilon) \oplus \epsilon^{\oplus 14} \oplus (-\epsilon),
\end{aligned} \tag{4.10}
$$

where $\epsilon = i\sigma_2$. The inequivalent automorphisms are then seen to be

$$\langle 2 \rangle, \ \ldots, \ \langle 8 \rangle, \ \langle 0, 1 \rangle, \ \langle 0, 15 \rangle, \ \langle 0, 16 \rangle. \tag{4.11}$$

That is, the only difference is that $\langle 0, 16 \rangle$ has split in two.

Not all automorphisms act as a $\mathbb{Z}_2$ in the spinor module. Note that the lattice of the trivial module is spanned by the simple roots $\alpha_i$ while that of the spinor module is spanned by the simple roots shifted by $\omega_{16}$. It is sufficient to consider a representative in the spinor module, $\Gamma_{\omega_{16}}$. If the symmetry in the spinor module is of order two, then by fusion $\Gamma_{2\omega_{16}}$ must be $\mathbb{Z}_2$-even. Let's compute the $\mathbb{Z}_2$ charge of $\Gamma_{2\omega_{16}}$:

$$e^{2\pi i (x, 2\omega_{16})} = e^{2\pi i \sum_{i=1}^{16} s_i (A^{-1})_{i,16}} = \begin{cases} +1, & n, n_1 + n_2 = 0 \bmod 2, \\ -1, & n, n_1 + n_2 = 1 \bmod 2. \end{cases} \tag{4.12}$$

This means that for $\langle n \rangle$ with odd $n$ and $\langle n_1, n_2 \rangle$ with odd $n_1 + n_2$, the symmetries acting on the spinor module have order four. We thus only focus on the symmetries $\langle n \rangle$ with even $n$ and $\langle n_1, n_2 \rangle$ with even $n_1 + n_2$, which are

$$\langle 2 \rangle, \ \langle 4 \rangle, \ \langle 6 \rangle, \ \langle 8 \rangle, \ \langle 0, 16 \rangle. \tag{4.13}$$

The above discussion only fixes the order-two automorphisms on the vertex operators in the spinor module up to an overall sign,

$$\Gamma_{\alpha+\omega_{16}} \to (-1)^\eta e^{2\pi i(x,\alpha+\omega_{16})}\Gamma_{\alpha+\omega_{16}}, \qquad \eta = 0,1, \tag{4.14}$$

and we are free to choose the sign. The $\eta$ dependence can be absorbed by shifting $x$ as $x \to x-\eta\omega_1$ since $e^{2\pi i(\eta\omega_1,\alpha+\omega_{16})} = (-1)^\eta$. Thus each order-two automorphism $\langle n \rangle$ and $\langle n_1, n_2 \rangle$ should be further labeled by $\eta$, distinguished by an overall minus sign on the vertex operators in the spinor module. We also need to include $\langle 0 \rangle_1$, which is a trivial (order-one) automorphism on $G$ but acts as $-1$ on the spinor module.

We can show that $\langle n \rangle_0$ and $\langle n \rangle_1$ are conjugate when $n \neq 0$, because $\frac{1}{2}\omega_n$ and $\frac{1}{2}\omega_n - \omega_1$ are conjugate. This is done by the lift of $\mathrm{diag}(-, +^{30}, -) \in \mathrm{SO}(32)$ to $\mathrm{Spin}(32)/\mathbb{Z}_2$, which is inner in the whole group but acts as an outer automorphism on the fixed subalgebra $\mathrm{SO}(2n)\times\mathrm{SO}(32-2n)$. Henceforth, we will only consider $\langle n \rangle_{\eta=0}$ for $n \neq 0$, and drop the subscript. In summary, we have the order-two automorphisms $\langle 0 \rangle_1$, $\langle n \rangle$ for $n = 2, 4, 6, 8$, and $\langle 0, 16 \rangle_{\eta=0,1}$.

Let us further determine the anomalies of the $\mathbb{Z}_2$ symmetries. Since we modified $x$, the formula (4.4) should be modified as

$$h = \frac{(x - \eta\omega_1, x - \eta\omega_1)}{2} = \frac{1}{8}\sum_{i,j=1}^{16} s_i s_j (A^{D_{16}})_{i,j}^{-1} - \frac{\eta}{2}\sum_{i=1}^{16} s_i (A^{D_{16}})_{i,1}^{-1} + \frac{\eta}{2}(A^{D_{16}})_{1,1}^{-1}. \tag{4.15}$$

Substituting (4.9) into the above, we have

$$\langle 0 \rangle_1 : \quad h_{\langle 0 \rangle_1} = 0 \bmod \tfrac{1}{2},$$

$$\langle n \rangle : \quad h_{\langle n \rangle} = \begin{cases} 0 \bmod \tfrac{1}{2}, & \langle n \rangle = \langle 4 \rangle, \langle 8 \rangle, \\ \tfrac{1}{4} \bmod \tfrac{1}{2}, & \langle n \rangle = \langle 2 \rangle, \langle 6 \rangle, \end{cases} \tag{4.16}$$

$$\langle n_1, n_2 \rangle_\eta : \quad h_{\langle n_1, n_2 \rangle_\eta} = \begin{cases} 0 \bmod \tfrac{1}{2}, & \langle n_1, n_2 \rangle_\eta = \langle 0, 16 \rangle_0, \\ \tfrac{1}{4} \bmod \tfrac{1}{2}, & \langle n_1, n_2 \rangle_\eta = \langle 0, 16 \rangle_1. \end{cases}$$

This implies that only $\langle 0 \rangle_1$, $\langle 4 \rangle$, $\langle 8 \rangle$ and $\langle 0, 16 \rangle_0$ are non-anomalous $\mathbb{Z}_2$ symmetries, while the others are all anomalous. One could then fermionize the $(\mathrm{Spin}(32)/\mathbb{Z}_2)_1$ chiral CFT by the above $\mathbb{Z}_2$ symmetries, whose invariant Lie subalgebras are $D_{16}$, $D_4 \times D_{12}$, $D_8 \times D_8'$, and $\mathrm{U}(1) \times A_{15}$, respectively.

# 5 Chiral fermionic CFTs with $c \leq 16$

Since we know all the chiral bosonic CFTs with $c \leq 16$ and have classified all the $\mathbb{Z}_2$ symmetries in these CFTs, we proceed to fermionize them to get the classification of chiral fermionic CFTs with $c \leq 16$. The fermionization operation is given in (2.2). In this section, we mainly work on

torus spacetime where (2.2) can be explicitly written as

$$\mathcal{Z}[F]^{\text{NS}}_{\text{NS}} = \frac{1}{2}(\mathcal{Z}[B] + \mathcal{Z}[B]^g + \mathcal{Z}[B]_g - \mathcal{Z}[B]^g_g),$$

$$\mathcal{Z}[F]^{\text{R}}_{\text{NS}} = \frac{1}{2}(\mathcal{Z}[B] + \mathcal{Z}[B]^g - \mathcal{Z}[B]_g + \mathcal{Z}[B]^g_g),$$

$$\mathcal{Z}[F]^{\text{NS}}_{\text{R}} = \frac{1}{2}(\mathcal{Z}[B] - \mathcal{Z}[B]^g + \mathcal{Z}[B]_g + \mathcal{Z}[B]^g_g),$$

$$\pm\mathcal{Z}[F]^{\text{R}}_{\text{R}} = \frac{1}{2}(\mathcal{Z}[B] - \mathcal{Z}[B]^g - \mathcal{Z}[B]_g - \mathcal{Z}[B]^g_g).$$

(5.1)

The superscript and subscript of $\mathcal{Z}[B]$ represent the insertion of the $g$ defect line along the spatial and time directions, respectively. The $\pm$ in the last equality represents the freedom of stacking with the $1+1$d SPT $(-1)^{\text{Arf}(q)}$ with spin structure $q$, and our convention (2.2) corresponds to the $+$ sign. For convenience, we introduce the notation $F[B, g]$ to denote the fermionization of a chiral bosonic CFT $B$ with respect to a non-anomalous $\mathbb{Z}_2$ symmetry generated by $g$.

The number $n_f$ of free Majorana-Weyl fermions in the chiral fermionic CFT $F$ is counted by the coefficient of $q^{\frac{1}{2}-\frac{c}{24}}$ of the partition function

$$\frac{1}{2}(\mathcal{Z}[F]^{\text{NS}}_{\text{NS}} - \mathcal{Z}[F]^{\text{R}}_{\text{NS}}) = \frac{1}{2}(\mathcal{Z}[B]_g - \mathcal{Z}[B]^g_g) = q^{\frac{1}{2}-\frac{c}{24}}(n_f + \mathcal{O}(q)).$$

(5.2)

Having extracted $n_f$, since a free fermion is a consistent CFT on its own, we can decouple $n_f$ free fermions from the theory to obtain a *seed* chiral fermionic CFT with central charge $c-\frac{n_f}{2}$ [GS88]. Here we define a seed CFT to have no free Majorana-Weyl fermions, i.e. $n_f = 0$. Below, we will explicitly check that the free fermions indeed decouple from the seed CFT in all of our examples.

In what follows, we will discuss the fermionization map without stacking an additional Arf invariant, which corresponds to a $+$ sign on the left hand side of the last equality in (5.1). The effect of the Arf invariant will be discussed along the way.

## 5.1   Fermionization of $(E_8)_1$

From Sec. 4.2, the $(E_8)_1$ theory has only one non-anomalous $\mathbb{Z}_2$ symmetry, generated by $a$. The invariant affine Lie subalgebra is $(D_8)_1$. Hence the twisted partition functions of $(E_8)_1$ are expressible in terms of the characters of $(D_8)_1$, i.e. $\chi_{\mathbf{1}}, \chi_{\mathbf{16}}, \chi_S$ and $\chi_C$. In App. A, we show that the states even and odd under $\mathbb{Z}^a_2$ sum to the $\chi^{D_8}_{\mathbf{1}}$ and $\chi^{D_8}_S$ of $(D_8)_1$, respectively, hence the untwisted partition function of $(E_8)_1$ is simply $\chi^{D_8}_{\mathbf{1}} + \chi^{D_8}_S$, and that with a $\mathbb{Z}^a_2$ line along the spatial direction $\mathcal{Z}[(E_8)_1]^a$ is $\chi^{D_8}_{\mathbf{1}} - \chi^{D_8}_S$. It is then straightforward to obtain the twisted partition functions: $\mathcal{Z}[(E_8)_1]_a$ by performing a modular $\mathcal{S}$ transformation, and $\mathcal{Z}[(E_8)_1]^a_a$ by an additional modular $\mathcal{T}$ transformation. The twisted partition functions are

$$\mathcal{Z}[(E_8)_1] = \chi^{D_8}_{\mathbf{1}} + \chi^{D_8}_S, \qquad \mathcal{Z}[(E_8)_1]^a = \chi^{D_8}_{\mathbf{1}} - \chi^{D_8}_S,$$

$$\mathcal{Z}[(E_8)_1]_a = \chi^{D_8}_C + \chi^{D_8}_{\mathbf{16}}, \qquad \mathcal{Z}[(E_8)_1]^a_a = \chi^{D_8}_C - \chi^{D_8}_{\mathbf{16}}.$$

(5.3)

The fully-refined characters of an affine Lie algebra depend not only on the torus modulus $\tau$ but also on the flavor fugacities. If we suppress the flavor fugacities, then [CS99, BKN21]

$$\mathcal{Z}[(E_8)_1] = \frac{E_4}{\eta^8}, \qquad \chi_{\mathbf{1}}^{D_8} = \frac{\theta_3^8 + \theta_4^8}{2\eta^8}, \qquad \chi_{\mathbf{16}}^{D_8} = \frac{\theta_3^8 - \theta_4^8}{2\eta^8}, \qquad \chi_S^{D_8} = \chi_C^{D_8} = \frac{\theta_2^8}{2\eta^8}, \quad (5.4)$$

which, by using $E_4 = (\theta_3^8 + \theta_4^8 + \theta_2^8)/2$, indeed satisfy (5.3). In particular, $\chi_S^{D_8}$ and $\chi_C^{D_8}$ are identical as $q$-series, but are different when the flavor fugacities are turned on. In particular, $\chi_S^{D_8}$ is $\mathbb{Z}_2^a$-odd, while $\chi_C^{D_8}$ is $\mathbb{Z}_2^a$-even. The difference between $\chi_S^{D_8}$ and $\chi_C^{D_8}$ is explained by the fact that they come from different lattice sums. The former comes from summing over the lattice $\langle \beta_1, ..., \beta_8 \rangle + \nu_8$, while the latter comes from summing over the lattice $\langle \beta_1, ..., \beta_8 \rangle + \nu_7$, and these respond differently to fugacities. See App. A for more on explicit formulas.

Using (5.1), the fermionized partition functions with various spin structures are straightforwardly determined,

$$\begin{aligned}
\mathcal{Z}[F[(E_8)_1; a]]_{\mathrm{NS}}^{\mathrm{NS}} &= \chi_{\mathbf{1}}^{D_8} + \chi_{\mathbf{16}}^{D_8}, &\qquad \mathcal{Z}[F[(E_8)_1; a]]_{\mathrm{NS}}^{\mathrm{R}} &= \chi_{\mathbf{1}}^{D_8} - \chi_{\mathbf{16}}^{D_8}, \\
\mathcal{Z}[F[(E_8)_1; a]]_{\mathrm{R}}^{\mathrm{NS}} &= \chi_S^{D_8} + \chi_C^{D_8}, &\qquad \mathcal{Z}[F[(E_8)_1; a]]_{\mathrm{R}}^{\mathrm{R}} &= \chi_S^{D_8} - \chi_C^{D_8}.
\end{aligned} \qquad (5.5)$$

The right hand sides are indeed the partition functions of 16 free Majorana-Weyl fermions. Indeed, the number $n_f = 16$ of free Majorana-Weyl fermions can be computed using (5.2) and the explicit $q$-series (5.4), as $\mathcal{Z}[(E_8)_1]_a - \mathcal{Z}[(E_8)_1]_a^a = 16q^{\frac{1}{6}} + \cdots$. In summary, we have

$$F[(E_8)_1; a] = 16\psi. \qquad (5.6)$$

Moreover, since stacking with an Arf invariant exchanges $\chi_S^{D_8}$ and $\chi_C^{D_8}$, $16\psi$ is invariant under stacking an Arf invariant, provided that we shift the background gauge fields for $\mathrm{Spin}(16)$ accordingly by an outer automorphism.

## 5.2 Fermionizations of $(E_8)_1 \times (E_8)_1$

### 5.2.1 By $\mathbb{Z}_2^{a_1}$

We proceed to fermionize the $(E_8)_1 \times (E_8)_1$ chiral bosonic CFT by the non-anomalous $\mathbb{Z}_2$ symmetry associated with the first $(E_8)_1$ component. This directly follows from the results in Sec. 5.1. Since the second $(E_8)_1$ component is completely decoupled, the twisted partition functions are simply given by multiplying (5.3) by $\chi_{\mathbf{1}'}^{E_8}$, and after fermionization, the partition functions are given by multiplying (5.5) by $\chi_{\mathbf{1}'}^{E_8}$ as well. In summary, we have

$$F[(E_8)_1 \times (E_8)_1; a_1] = 16\psi \times (E_8)_1. \qquad (5.7)$$

The seed theory $(E_8)_1$ is distinct from $(E_8)_1 \times \mathrm{Arf}$, as this holds true for any bosonic theory.

### 5.2.2 By $\mathbb{Z}_2^{a_1 a_2}$

The twisted partition functions of $(E_8)_1 \times (E_8)_1$ by $\mathbb{Z}_2^{a_1 a_2}$ are simply obtained by squaring the twisted partition functions of a single copy of $(E_8)_1$ by $\mathbb{Z}_2^a$. Concretely,

$$
\begin{aligned}
\mathcal{Z}[(E_8)_1 \times (E_8)_1] \quad &= (\chi_{\mathbf{1}}^{D_8} + \chi_S^{D_8})(\chi_{\mathbf{1'}}^{D_8} + \chi_{S'}^{D_8}), \\
\mathcal{Z}[(E_8)_1 \times (E_8)_1]^{a_1 a_2} &= (\chi_{\mathbf{1}}^{D_8} - \chi_S^{D_8})(\chi_{\mathbf{1'}}^{D_8} - \chi_{S'}^{D_8}), \\
\mathcal{Z}[(E_8)_1 \times (E_8)_1]_{a_1 a_2} &= (\chi_C^{D_8} + \chi_{\mathbf{16}}^{D_8})(\chi_{C'}^{D_8} + \chi_{\mathbf{16'}}^{D_8}), \\
\mathcal{Z}[(E_8)_1 \times (E_8)_1]_{a_1 a_2}^{a_1 a_2} &= (\chi_C^{D_8} - \chi_{\mathbf{16}}^{D_8})(\chi_{C'}^{D_8} - \chi_{\mathbf{16'}}^{D_8}).
\end{aligned}
\tag{5.8}
$$

Under fermionization,

$$
\begin{aligned}
\mathcal{Z}[F[(E_8)_1 \times (E_8)_1; a_1 a_2]]_{\mathrm{NS}}^{\mathrm{NS}} &= \chi_{\mathbf{1}}^{D_8}\chi_{\mathbf{1'}}^{D_8} + \chi_S^{D_8}\chi_{S'}^{D_8} + \chi_{\mathbf{16}}^{D_8}\chi_{C'}^{D_8} + \chi_C^{D_8}\chi_{\mathbf{16'}}^{D_8}, \\
\mathcal{Z}[F[(E_8)_1 \times (E_8)_1; a_1 a_2]]_{\mathrm{NS}}^{\mathrm{R}} &= \chi_{\mathbf{1}}^{D_8}\chi_{\mathbf{1'}}^{D_8} + \chi_S^{D_8}\chi_{S'}^{D_8} - \chi_{\mathbf{16}}^{D_8}\chi_{C'}^{D_8} - \chi_C^{D_8}\chi_{\mathbf{16'}}^{D_8}, \\
\mathcal{Z}[F[(E_8)_1 \times (E_8)_1; a_1 a_2]]_{\mathrm{R}}^{\mathrm{NS}} &= \chi_{\mathbf{1}}^{D_8}\chi_{S'}^{D_8} + \chi_{\mathbf{1'}}^{D_8}\chi_S^{D_8} + \chi_{\mathbf{16}}^{D_8}\chi_{\mathbf{16'}}^{D_8} + \chi_C^{D_8}\chi_{C'}^{D_8}, \\
\mathcal{Z}[F[(E_8)_1 \times (E_8)_1; a_1 a_2]]_{\mathrm{R}}^{\mathrm{R}} &= \chi_{\mathbf{1}}^{D_8}\chi_{S'}^{D_8} + \chi_{\mathbf{1'}}^{D_8}\chi_S^{D_8} - \chi_{\mathbf{16}}^{D_8}\chi_{\mathbf{16'}}^{D_8} - \chi_C^{D_8}\chi_{C'}^{D_8}.
\end{aligned}
\tag{5.9}
$$

This fermionic theory $F[(E_8)_1 \times (E_8)_1; a_1 a_2]$ does not contain any free Majorana-Weyl fermions, as can be seen by checking that the $q$ expansion has a vanishing coefficient of $q^{\frac{1}{2} - \frac{c}{24}} = q^{-\frac{1}{6}}$. Thus this is a seed chiral fermionic CFT, which we denote as $\overline{(D_8)_1 \times (D_8)_1}$.[20] In summary, we have

$$
F[(E_8)_1 \times (E_8)_1; a_1 a_2] = \overline{(D_8)_1 \times (D_8)_1}.
\tag{5.10}
$$

Unlike in the $16\psi$ case, this time flipping the sign of the RR partition function cannot be compensated by a shift in the background gauge fields. Thus $\overline{(D_8)_1 \times (D_8)_1}$ and $\overline{(D_8)_1 \times (D_8)_1} \times \mathrm{Arf}$ are distinct theories.

### 5.2.3 By $\mathbb{Z}_2^{b_1 b_2}$

The twisted partition functions of $(E_8)_1 \times (E_8)_1$ by $\mathbb{Z}_2^{b_1 b_2}$ are simply obtained by squaring the twisted partition functions of a single copy of $(E_8)_1$ by $\mathbb{Z}_2^b$. In particular, $\mathcal{Z}[(E_8)_1]^b$ has been discussed in (A.14). By a modular $\mathcal{S}$ transformation, we can derive $\mathcal{Z}[(E_8)_1]_b$. Further applying a modular $\mathcal{T}$ transformation, we have $\mathcal{Z}[(E_8)_1]_b^b$. Concretely,

$$
\begin{aligned}
\mathcal{Z}[(E_8)_1] &= \chi_{\mathbf{1}}^{A_1}\chi_{\mathbf{1}}^{E_7} + \chi_{\mathbf{2}}^{A_1}\chi_{\mathbf{56}}^{E_7}, &\qquad \mathcal{Z}[(E_8)_1]^b &= \chi_{\mathbf{1}}^{A_1}\chi_{\mathbf{1}}^{E_7} - \chi_{\mathbf{2}}^{A_1}\chi_{\mathbf{56}}^{E_7}, \\
\mathcal{Z}[(E_8)_1]_b &= \chi_{\mathbf{1}}^{A_1}\chi_{\mathbf{56}}^{E_7} + \chi_{\mathbf{2}}^{A_1}\chi_{\mathbf{1}}^{E_7}, &\qquad \mathcal{Z}[(E_8)_1]_b^b &= -i\chi_{\mathbf{1}}^{A_1}\chi_{\mathbf{56}}^{E_7} + i\chi_{\mathbf{2}}^{A_1}\chi_{\mathbf{1}}^{E_7}.
\end{aligned}
\tag{5.11}
$$

Squaring each twisted partition function yields the partition function twisted by $b_1 b_2$,

$$
\begin{aligned}
\mathcal{Z}[(E_8)_1 \times (E_8)_1] &= (\chi_{\mathbf{1}}^{A_1}\chi_{\mathbf{1}}^{E_7} + \chi_{\mathbf{2}}^{A_1}\chi_{\mathbf{56}}^{E_7})(\chi_{\mathbf{1'}}^{A_1}\chi_{\mathbf{1'}}^{E_7} + \chi_{\mathbf{2'}}^{A_1}\chi_{\mathbf{56'}}^{E_7}), \\
\mathcal{Z}[(E_8)_1 \times (E_8)_1]^{b_1 b_2} &= (\chi_{\mathbf{1}}^{A_1}\chi_{\mathbf{1}}^{E_7} - \chi_{\mathbf{2}}^{A_1}\chi_{\mathbf{56}}^{E_7})(\chi_{\mathbf{1'}}^{A_1}\chi_{\mathbf{1'}}^{E_7} - \chi_{\mathbf{2'}}^{A_1}\chi_{\mathbf{56'}}^{E_7}), \\
\mathcal{Z}[(E_8)_1 \times (E_8)_1]_{b_1 b_2} &= (\chi_{\mathbf{1}}^{A_1}\chi_{\mathbf{56}}^{E_7} + \chi_{\mathbf{2}}^{A_1}\chi_{\mathbf{1}}^{E_7})(\chi_{\mathbf{1'}}^{A_1}\chi_{\mathbf{56'}}^{E_7} + \chi_{\mathbf{2'}}^{A_1}\chi_{\mathbf{1'}}^{E_7}), \\
\mathcal{Z}[(E_8)_1 \times (E_8)_1]_{b_1 b_2}^{b_1 b_2} &= -(\chi_{\mathbf{1}}^{A_1}\chi_{\mathbf{56}}^{E_7} - \chi_{\mathbf{2}}^{A_1}\chi_{\mathbf{1}}^{E_7})(\chi_{\mathbf{1'}}^{A_1}\chi_{\mathbf{56'}}^{E_7} - \chi_{\mathbf{2'}}^{A_1}\chi_{\mathbf{1'}}^{E_7}).
\end{aligned}
\tag{5.12}
$$

---

[20]Here and in the following, we use $\overline{G_k}$ to denote the modular invariant chiral CFT extending the $G_k$ chiral algebra to emphasize that the vacuum representation of $G_k$ alone is not a modular invariant chiral CFT, but the specific combination of characters such as (5.9) is.

Under fermionization,

$$\mathcal{Z}[F[(E_8)_1 \times (E_8)_1; b_1 b_2]]_{\text{NS}}^{\text{NS}} = (\chi_{\mathbf{1}}^{A_1}\chi_{\mathbf{1'}}^{A_1} + \chi_{\mathbf{2}}^{A_1}\chi_{\mathbf{2'}}^{A_1})(\chi_{\mathbf{1}}^{E_7}\chi_{\mathbf{1'}}^{E_7} + \chi_{\mathbf{56}}^{E_7}\chi_{\mathbf{56'}}^{E_7}),$$
$$\mathcal{Z}[F[(E_8)_1 \times (E_8)_1; b_1 b_2]]_{\text{NS}}^{\text{R}} = (\chi_{\mathbf{1}}^{A_1}\chi_{\mathbf{1'}}^{A_1} - \chi_{\mathbf{2}}^{A_1}\chi_{\mathbf{2'}}^{A_1})(\chi_{\mathbf{1}}^{E_7}\chi_{\mathbf{1'}}^{E_7} - \chi_{\mathbf{56}}^{E_7}\chi_{\mathbf{56'}}^{E_7}),$$
$$\mathcal{Z}[F[(E_8)_1 \times (E_8)_1; b_1 b_2]]_{\text{R}}^{\text{NS}} = (\chi_{\mathbf{2}}^{A_1}\chi_{\mathbf{1'}}^{A_1} + \chi_{\mathbf{1}}^{A_1}\chi_{\mathbf{2'}}^{A_1})(\chi_{\mathbf{1}}^{E_7}\chi_{\mathbf{56'}}^{E_7} + \chi_{\mathbf{56}}^{E_7}\chi_{\mathbf{1'}}^{E_7}),$$
$$\mathcal{Z}[F[(E_8)_1 \times (E_8)_1; b_1 b_2]]_{\text{R}}^{\text{R}} = (\chi_{\mathbf{2}}^{A_1}\chi_{\mathbf{1'}}^{A_1} - \chi_{\mathbf{1}}^{A_1}\chi_{\mathbf{2'}}^{A_1})(\chi_{\mathbf{1}}^{E_7}\chi_{\mathbf{56'}}^{E_7} - \chi_{\mathbf{56}}^{E_7}\chi_{\mathbf{1'}}^{E_7}). \tag{5.13}$$

The twisted partition functions factorize, where the first components are those of four free Majorana-Weyl fermions. Indeed, the $q$-series of the first components are

$$\chi_{\mathbf{1}}^{A_1}\chi_{\mathbf{1'}}^{A_1} + \chi_{\mathbf{2}}^{A_1}\chi_{\mathbf{2'}}^{A_1} = \frac{\theta_3^2}{\eta^2}, \qquad \chi_{\mathbf{1}}^{A_1}\chi_{\mathbf{1'}}^{A_1} - \chi_{\mathbf{2}}^{A_1}\chi_{\mathbf{2'}}^{A_1} = \frac{\theta_4^2}{\eta^2},$$
$$\chi_{\mathbf{2}}^{A_1}\chi_{\mathbf{1'}}^{A_1} + \chi_{\mathbf{1}}^{A_1}\chi_{\mathbf{2'}}^{A_1} = \frac{\theta_2^2}{\eta^2}, \qquad \chi_{\mathbf{2}}^{A_1}\chi_{\mathbf{1'}}^{A_1} - \chi_{\mathbf{1}}^{A_1}\chi_{\mathbf{2'}}^{A_1} = 0, \tag{5.14}$$

which coincide with those of four free fermions. In summary, we have

$$F[(E_8)_1 \times (E_8)_1; b_1 b_2] = 4\psi \times \overline{(E_7)_1 \times (E_7)_1}. \tag{5.15}$$

It can be explicitly seen that $\overline{(E_7)_1 \times (E_7)_1}$ can absorb an Arf, since this operation can be compensated for by exchanging the two $(E_7)_1$ factors.

### 5.2.4 By $\mathbb{Z}_2^\sigma$

The twisted partition functions of $n$ copies of a purely chiral CFT $T$ under the $\mathbb{Z}_n$ cyclic permutation symmetry are[21]

$$\mathcal{Z}[T^{\otimes n}](\tau) = \mathcal{Z}[T](\tau)^n, \qquad \mathcal{Z}[T^{\otimes n}]^\sigma(\tau) = \mathcal{Z}[T](n\tau),$$
$$\mathcal{Z}[T^{\otimes n}]_\sigma(\tau) = \mathcal{Z}[T](\tfrac{\tau}{n}), \qquad \mathcal{Z}[T^{\otimes n}]_\sigma^\sigma(\tau) = \mathcal{Z}[T](\tfrac{\tau+1}{n}) e^{-2\pi i \frac{c}{24}}. \tag{5.16}$$

Since the theory of interest is $(E_8)_1 \times (E_8)_1$, this corresponds to $n = 2$. As the untwisted partition function for a single copy of $(E_8)_1$ is $E_4(\tau)/\eta(\tau)^8$, we obtain the twisted partition functions for $(E_8)_1 \times (E_8)_1$,

$$\mathcal{Z}[(E_8)_1 \times (E_8)_1] = \frac{E_4(\tau)^2}{\eta(\tau)^{16}}, \qquad \mathcal{Z}[(E_8)_1 \times (E_8)_1]^\sigma = \frac{E_4(2\tau)}{\eta(2\tau)^8},$$
$$\mathcal{Z}[(E_8)_1 \times (E_8)_1]_\sigma = \frac{E_4(\frac{\tau}{2})}{\eta(\frac{\tau}{2})^8}, \qquad \mathcal{Z}[(E_8)_1 \times (E_8)_1]_\sigma^\sigma = \frac{E_4(\frac{\tau+1}{2})}{\eta(\frac{\tau+1}{2})^8} e^{-2\pi i/3}. \tag{5.17}$$

From the coefficient of $q^{\frac{1}{2} - \frac{c}{24}} = q^{-\frac{1}{6}}$ in $\frac{1}{2}(\mathcal{Z}[(E_8)_1 \times (E_8)_1]_\sigma - \mathcal{Z}[(E_8)_1 \times (E_8)_1]_\sigma^\sigma)$, we obtain $n_f = 1$. This means that the fermionization of $(E_8)_1 \times (E_8)_1$ by $\mathbb{Z}_2^\sigma$ contains one free Majorana-Weyl fermion. Moreover, it is obvious that the affine symmetry left unbroken by $\mathbb{Z}_2^\sigma$ contains

---

[21]See [AKL22] for a discussion of the permutation anomaly using these formulas.

$(E_8)_2$. Finally, note that the standard coset relation [DFMS97, Chapter 18][22]

$$\frac{(E_8)_1 \times (E_8)_1}{(E_8)_2} = \text{Vir}_{c=1/2} \tag{5.18}$$

implies that the $\mathbb{Z}_2^\sigma$ twisted partition functions of $(E_8)_1 \times (E_8)_1$ can be decomposed into the characters of $(E_8)_2$, i.e.

$$\chi_{2\omega_0}^{E_8} := \chi_{\mathbf{1}}^{E_8}, \qquad \chi_{\omega_1}^{E_8} := \chi_{\mathbf{248}}^{E_8}, \qquad \chi_{\omega_7}^{E_8} := \chi_{\mathbf{3875}}^{E_8}, \tag{5.19}$$

and the $c = 1/2$ Virasoro characters associated to the Ising CFT, i.e. $\chi_0^{\text{Vir}}$, $\chi_{1/2}^{\text{Vir}}$, $\chi_{1/16}^{\text{Vir}}$. In terms of these characters, the twisted partition functions (5.17) are

$$
\begin{aligned}
\mathcal{Z}[(E_8)_1 \times (E_8)_1] &= \chi_0^{\text{Vir}}\chi_{\mathbf{1}}^{E_8} + \chi_{1/2}^{\text{Vir}}\chi_{\mathbf{3875}}^{E_8} + \chi_{1/16}^{\text{Vir}}\chi_{\mathbf{248}}^{E_8}, \\
\mathcal{Z}[(E_8)_1 \times (E_8)_1]^\sigma &= \chi_0^{\text{Vir}}\chi_{\mathbf{1}}^{E_8} + \chi_{1/2}^{\text{Vir}}\chi_{\mathbf{3875}}^{E_8} - \chi_{1/16}^{\text{Vir}}\chi_{\mathbf{248}}^{E_8}, \\
\mathcal{Z}[(E_8)_1 \times (E_8)_1]_\sigma &= \chi_{1/2}^{\text{Vir}}\chi_{\mathbf{1}}^{E_8} + \chi_0^{\text{Vir}}\chi_{\mathbf{3875}}^{E_8} + \chi_{1/16}^{\text{Vir}}\chi_{\mathbf{248}}^{E_8}, \\
\mathcal{Z}[(E_8)_1 \times (E_8)_1]_\sigma^\sigma &= -\chi_{1/2}^{\text{Vir}}\chi_{\mathbf{1}}^{E_8} - \chi_0^{\text{Vir}}\chi_{\mathbf{3875}}^{E_8} + \chi_{1/16}^{\text{Vir}}\chi_{\mathbf{248}}^{E_8}.
\end{aligned}
\tag{5.20}
$$

Under fermionization,

$$
\begin{aligned}
\mathcal{Z}[F[(E_8)_1 \times (E_8)_1; \sigma]]_{\text{NS}}^{\text{NS}} &= (\chi_0^{\text{Vir}} + \chi_{1/2}^{\text{Vir}})(\chi_{\mathbf{1}}^{E_8} + \chi_{\mathbf{3875}}^{E_8}), \\
\mathcal{Z}[F[(E_8)_1 \times (E_8)_1; \sigma]]_{\text{NS}}^{\text{R}} &= (\chi_0^{\text{Vir}} - \chi_{1/2}^{\text{Vir}})(\chi_{\mathbf{1}}^{E_8} - \chi_{\mathbf{3875}}^{E_8}), \\
\mathcal{Z}[F[(E_8)_1 \times (E_8)_1; \sigma]]_{\text{R}}^{\text{NS}} &= \sqrt{2}\chi_{1/16}^{\text{Vir}} \cdot \sqrt{2}\chi_{\mathbf{248}}^{E_8}, \\
\mathcal{Z}[F[(E_8)_1 \times (E_8)_1; \sigma]]_{\text{R}}^{\text{R}} &= 0.
\end{aligned}
\tag{5.21}
$$

Note that the partition functions on the right hand side factorize into the Virasoro and $E_8$ parts.

The first factors

$$\chi_0^{\text{Vir}} + \chi_{1/2}^{\text{Vir}} = \sqrt{\frac{\theta_3}{\eta}}, \qquad \chi_0^{\text{Vir}} - \chi_{1/2}^{\text{Vir}} = \sqrt{\frac{\theta_4}{\eta}}, \qquad \sqrt{2}\chi_{1/16}^{\text{Vir}} = \sqrt{\frac{\theta_2}{\eta}}. \tag{5.22}$$

are precisely the partition functions of a single free Majorana-Weyl fermion, as expected. The characters $\chi_{\mathbf{1}}^{E_8}, \chi_{\mathbf{3875}}^{E_8}, \chi_{\mathbf{248}}^{E_8}$ for the $(E_8)_2$ seed chiral fermionic CFT transform in the conjugate modular representation of the Virasoro characters $\chi_0^{\text{Vir}}, \chi_{1/2}^{\text{Vir}}, \chi_{1/16}^{\text{Vir}}$ for the free Majorana-Weyl fermion theory. In summary, we have

$$F[(E_8)_1 \times (E_8)_1; \sigma] = \psi \times \overline{(E_8)_2}. \tag{5.23}$$

The factors of $\sqrt{2}$ in the third line of (5.21) can be thought of as the 'quantum dimension' of the Hilbert space acted on by a single real fermion zero mode, or by any system with half-integral $c$ in the R-sector. By combining the $\psi$ part and the $\overline{(E_8)_2}$ part, we have an integer dimension.

---

[22]The coset construction often refers to the quotient of fully modular invariant WZW models. However, for consistency with the rest of the paper, here we define the coset in terms of commutant relations of chiral algebras, e.g. the commutant subalgebra of $(E_8)_2 \subset (E_8)_1 \times (E_8)_1$ is the Virasoro algebra at $c = 1/2$, which is the chiral algebra of the Ising CFT.

Note that, unlike the previous cases as e.g. in Sec. 5.1 where the RR partition function can be made non-zero by turning on fugacities, here $F[(E_8)_1 \times (E_8)_1; \sigma]_{\text{R}}^{\text{R}}$ identically vanishes. The fact that it vanishes comes from the single fermion zero mode in one Majorana-Weyl fermion. This leaves the RR partition function of $\overline{(E_8)_2}$ undetermined. However, it can be shown that it also identically vanishes. This is because it would have to transform into itself up to a phase under $\mathcal{S}$ and $\mathcal{T}$ transformations, but no nonzero combination of the characters achieves this. Hence the $\overline{(E_8)_2}$ theory can absorb an Arf invariant.

## 5.3 Fermionizations of $(\text{Spin}(32)/\mathbb{Z}_2)_1$

### 5.3.1 By $\mathbb{Z}_2^{\langle 0 \rangle_1}$

The $\mathbb{Z}_2^{\langle 0 \rangle_1}$ symmetry leaves the entire $D_{16}$ Lie algebra invariant, so the twisted partition functions should be expressed in terms of characters of $D_{16}$, i.e. $\chi_{\mathbf{1}}^{D_{16}}, \chi_{S}^{D_{16}}, \chi_{\mathbf{16}}^{D_{16}}, \chi_{C}^{D_{16}}$. From Sec. 4.2, under $\langle 0 \rangle_1$, the character $\chi_{\mathbf{1}}^{D_{16}}$ is invariant, while $\chi_{S}^{D_{16}}$ acquires a minus sign. By further using the modular $\mathcal{S}$ and $\mathcal{T}$ transformations, we can get the remaining twisted partition functions. Denoting by $g$ the generator of $\langle 0 \rangle_1$, we have

$$
\begin{aligned}
\mathcal{Z}[(\text{Spin}(32)/\mathbb{Z}_2)_1] &= \chi_{\mathbf{1}}^{D_{16}} + \chi_{S}^{D_{16}}, & \mathcal{Z}[(\text{Spin}(32)/\mathbb{Z}_2)_1]^g &= \chi_{\mathbf{1}}^{D_{16}} - \chi_{S}^{D_{16}}, \\
\mathcal{Z}[(\text{Spin}(32)/\mathbb{Z}_2)_1]_g &= \chi_{C}^{D_{16}} + \chi_{\mathbf{32}}^{D_{16}}, & \mathcal{Z}[(\text{Spin}(32)/\mathbb{Z}_2)_1]_g^g &= \chi_{C}^{D_{16}} - \chi_{\mathbf{32}}^{D_{16}}.
\end{aligned}
\tag{5.24}
$$

Under fermionization,

$$
\begin{aligned}
\mathcal{Z}[F[(\text{Spin}(32)/\mathbb{Z}_2)_1; \langle 0 \rangle_1]]_{\text{NS}}^{\text{NS}} &= \chi_{\mathbf{1}}^{D_{16}} + \chi_{\mathbf{32}}^{D_{16}}, \\
\mathcal{Z}[F[(\text{Spin}(32)/\mathbb{Z}_2)_1; \langle 0 \rangle_1]]_{\text{NS}}^{\text{R}} &= \chi_{\mathbf{1}}^{D_{16}} - \chi_{\mathbf{32}}^{D_{16}}, \\
\mathcal{Z}[F[(\text{Spin}(32)/\mathbb{Z}_2)_1; \langle 0 \rangle_1]]_{\text{R}}^{\text{NS}} &= \chi_{S}^{D_{16}} + \chi_{C}^{D_{16}}, \\
\mathcal{Z}[F[(\text{Spin}(32)/\mathbb{Z}_2)_1; \langle 0 \rangle_1]]_{\text{R}}^{\text{R}} &= \chi_{S}^{D_{16}} - \chi_{C}^{D_{16}}.
\end{aligned}
\tag{5.25}
$$

From (5.2) we find $n_f = 32$. Indeed, the right hand side is the set of partition functions of 32 free Majorana-Weyl fermions. In summary,

$$
F[(\text{Spin}(32)/\mathbb{Z}_2)_1; \langle 0 \rangle_1] = 32\psi.
\tag{5.26}
$$

### 5.3.2 By $\mathbb{Z}_2^{\langle 4 \rangle}$

The $\mathbb{Z}_2^{\langle 4 \rangle}$ symmetry leaves the $D_4 \times D_{12}$ Lie subalgebra invariant, so the twisted partition functions should be expressible in terms of the characters of $D_4$ and $D_{12}$. Denote by $g$ the generator of $\langle 4 \rangle$; the decomposition of $\mathcal{Z}[(\text{Spin}(32)/\mathbb{Z}_2)_1]$ and $\mathcal{Z}[(\text{Spin}(32)/\mathbb{Z}_2)_1]^g$ is discussed in App. B. By further using the modular $\mathcal{S}$ and $\mathcal{T}$ transformations, we can get the remaining twisted partition functions,

$$
\begin{aligned}
\mathcal{Z}[(\text{Spin}(32)/\mathbb{Z}_2)_1] &= \chi_{\mathbf{1}}^{D_4}\chi_{\mathbf{1}}^{D_{12}} + \chi_{S}^{D_4}\chi_{S}^{D_{12}} + \chi_{\mathbf{8}}^{D_4}\chi_{\mathbf{24}}^{D_{12}} + \chi_{C}^{D_4}\chi_{C}^{D_{12}}, \\
\mathcal{Z}[(\text{Spin}(32)/\mathbb{Z}_2)_1]^g &= \chi_{\mathbf{1}}^{D_4}\chi_{\mathbf{1}}^{D_{12}} + \chi_{S}^{D_4}\chi_{S}^{D_{12}} - \chi_{\mathbf{8}}^{D_4}\chi_{\mathbf{24}}^{D_{12}} - \chi_{C}^{D_4}\chi_{C}^{D_{12}}, \\
\mathcal{Z}[(\text{Spin}(32)/\mathbb{Z}_2)_1]_g &= \chi_{\mathbf{1}}^{D_4}\chi_{S}^{D_{12}} + \chi_{S}^{D_4}\chi_{\mathbf{1}}^{D_{12}} + \chi_{C}^{D_4}\chi_{\mathbf{24}}^{D_{12}} + \chi_{\mathbf{8}}^{D_4}\chi_{C}^{D_{12}}, \\
\mathcal{Z}[(\text{Spin}(32)/\mathbb{Z}_2)_1]_g^g &= -\chi_{\mathbf{1}}^{D_4}\chi_{S}^{D_{12}} - \chi_{S}^{D_4}\chi_{\mathbf{1}}^{D_{12}} + \chi_{C}^{D_4}\chi_{\mathbf{24}}^{D_{12}} + \chi_{\mathbf{8}}^{D_4}\chi_{C}^{D_{12}}.
\end{aligned}
\tag{5.27}
$$

Then, under fermionization, we have

$$
\begin{aligned}
\mathcal{Z}[F[(\mathrm{Spin}(32)/\mathbb{Z}_2)_1; \langle 4 \rangle]]_{\mathrm{NS}}^{\mathrm{NS}} &= (\chi_{\mathbf{1}}^{D_4} + \chi_S^{D_4})(\chi_{\mathbf{1}}^{D_{12}} + \chi_S^{D_{12}}), \\
\mathcal{Z}[F[(\mathrm{Spin}(32)/\mathbb{Z}_2)_1; \langle 4 \rangle]]_{\mathrm{NS}}^{\mathrm{R}} &= (\chi_{\mathbf{1}}^{D_4} - \chi_S^{D_4})(\chi_{\mathbf{1}}^{D_{12}} - \chi_S^{D_{12}}), \\
\mathcal{Z}[F[(\mathrm{Spin}(32)/\mathbb{Z}_2)_1; \langle 4 \rangle]]_{\mathrm{R}}^{\mathrm{NS}} &= (\chi_{\mathbf{8}}^{D_4} + \chi_C^{D_4})(\chi_{\mathbf{24}}^{D_{12}} + \chi_C^{D_{12}}), \\
\mathcal{Z}[F[(\mathrm{Spin}(32)/\mathbb{Z}_2)_1; \langle 4 \rangle]]_{\mathrm{R}}^{\mathrm{R}} &= (\chi_{\mathbf{8}}^{D_4} - \chi_C^{D_4})(\chi_{\mathbf{24}}^{D_{12}} - \chi_C^{D_{12}}).
\end{aligned}
\tag{5.28}
$$

The first factors on the right hand side are (in terms of $q$-series)

$$
\chi_{\mathbf{1}}^{D_4} + \chi_S^{D_4} = \frac{\theta_3^4}{\eta^4}, \qquad \chi_{\mathbf{1}}^{D_4} - \chi_S^{D_4} = \frac{\theta_4^4}{\eta^4}, \qquad \chi_{\mathbf{8}}^{D_4} + \chi_C^{D_4} = \frac{\theta_2^4}{\eta^4}, \qquad \chi_{\mathbf{8}}^{D_4} - \chi_C^{D_4} = 0, \tag{5.29}
$$

which are indeed the partition functions of 8 free Majorana-Weyl fermions. On the other hand, the second factors on the right hand side of (5.28) are

$$
\begin{aligned}
\chi_{\mathbf{1}}^{D_{12}} + \chi_S^{D_{12}} &= \frac{\theta_3^{12} + \theta_4^{12} + \theta_2^{12}}{2\eta^{12}}, & \chi_{\mathbf{1}}^{D_{12}} - \chi_S^{D_{12}} &= \frac{\theta_3^{12} + \theta_4^{12} - \theta_2^{12}}{2\eta^{12}}, \\
\chi_{\mathbf{24}}^{D_{12}} + \chi_C^{D_{12}} &= \frac{\theta_3^{12} - \theta_4^{12} + \theta_2^{12}}{2\eta^{12}}, & \chi_{\mathbf{24}}^{D_{12}} - \chi_C^{D_{12}} &= 24.
\end{aligned}
\tag{5.30}
$$

We denote this theory by $\overline{(D_{12})_1}$, to distinguish it from the chiral algebra $(D_{12})_1$. This theory is isomorphic to Supermoonshine [Dun05, DMC14]. See also a ternary code construction in [GJF18]. In summary, we find

$$
F[(\mathrm{Spin}(32)/\mathbb{Z}_2)_1; \langle 4 \rangle] = 4\psi \times \overline{(D_{12})_1}. \tag{5.31}
$$

The theories $\overline{(D_{12})_1}$ and $\overline{(D_{12})_1} \times \mathrm{Arf}$ are distinct.

### 5.3.3 By $\mathbb{Z}_2^{\langle 8 \rangle}$

The discussion here goes in parallel with the immediately preceding case. The invariant Lie subalgebra is $D_8 \times D_8$. With $g$ denoting the generator of $\mathbb{Z}_2^{\langle 8 \rangle}$, the twisted partition functions are

$$
\begin{aligned}
\mathcal{Z}[(\mathrm{Spin}(32)/\mathbb{Z}_2)_1] &= \chi_{\mathbf{1}}^{D_8}\chi_{\mathbf{1}'}^{D_8} + \chi_S^{D_8}\chi_{S'}^{D_8} + \chi_{\mathbf{16}}^{D_8}\chi_{\mathbf{16}'}^{D_8} + \chi_C^{D_8}\chi_{C'}^{D_8}, \\
\mathcal{Z}[(\mathrm{Spin}(32)/\mathbb{Z}_2)_1]^g &= \chi_{\mathbf{1}}^{D_8}\chi_{\mathbf{1}'}^{D_8} + \chi_S^{D_8}\chi_{S'}^{D_8} - \chi_{\mathbf{16}}^{D_8}\chi_{\mathbf{16}'}^{D_8} - \chi_C^{D_8}\chi_{C'}^{D_8}, \\
\mathcal{Z}[(\mathrm{Spin}(32)/\mathbb{Z}_2)_1]_g &= \chi_{\mathbf{1}}^{D_8}\chi_{S'}^{D_8} + \chi_S^{D_8}\chi_{\mathbf{1}'}^{D_8} + \chi_C^{D_8}\chi_{\mathbf{16}'}^{D_8} + \chi_{\mathbf{16}}^{D_8}\chi_{C'}^{D_8}, \\
\mathcal{Z}[(\mathrm{Spin}(32)/\mathbb{Z}_2)_1]_g^g &= \chi_{\mathbf{1}}^{D_8}\chi_{S'}^{D_8} + \chi_S^{D_8}\chi_{\mathbf{1}'}^{D_8} - \chi_C^{D_8}\chi_{\mathbf{16}'}^{D_8} - \chi_{\mathbf{16}}^{D_8}\chi_{C'}^{D_8}.
\end{aligned}
\tag{5.32}
$$

Under fermionization, we have

$$
\begin{aligned}
\mathcal{Z}[F[(\mathrm{Spin}(32)/\mathbb{Z}_2)_1; \langle 8 \rangle]]_{\mathrm{NS}}^{\mathrm{NS}} &= \chi_{\mathbf{1}}^{D_8}\chi_{\mathbf{1}'}^{D_8} + \chi_S^{D_8}\chi_{S'}^{D_8} + \chi_{\mathbf{16}}^{D_8}\chi_{C'}^{D_8} + \chi_C^{D_8}\chi_{\mathbf{16}'}^{D_8}, \\
\mathcal{Z}[F[(\mathrm{Spin}(32)/\mathbb{Z}_2)_1; \langle 8 \rangle]]_{\mathrm{NS}}^{\mathrm{R}} &= \chi_{\mathbf{1}}^{D_8}\chi_{\mathbf{1}'}^{D_8} + \chi_S^{D_8}\chi_{S'}^{D_8} - \chi_{\mathbf{16}}^{D_8}\chi_{C'}^{D_8} - \chi_C^{D_8}\chi_{\mathbf{16}'}^{D_8}, \\
\mathcal{Z}[F[(\mathrm{Spin}(32)/\mathbb{Z}_2)_1; \langle 8 \rangle]]_{\mathrm{R}}^{\mathrm{NS}} &= \chi_{\mathbf{1}}^{D_8}\chi_{S'}^{D_8} + \chi_{\mathbf{1}'}^{D_8}\chi_S^{D_8} + \chi_{\mathbf{16}}^{D_8}\chi_{\mathbf{16}'}^{D_8} + \chi_C^{D_8}\chi_{C'}^{D_8}, \\
\mathcal{Z}[F[(\mathrm{Spin}(32)/\mathbb{Z}_2)_1; \langle 8 \rangle]]_{\mathrm{R}}^{\mathrm{R}} &= -\chi_{\mathbf{1}}^{D_8}\chi_{S'}^{D_8} - \chi_{\mathbf{1}'}^{D_8}\chi_S^{D_8} + \chi_{\mathbf{16}}^{D_8}\chi_{\mathbf{16}'}^{D_8} + \chi_C^{D_8}\chi_{C'}^{D_8}.
\end{aligned}
\tag{5.33}
$$

In particular, the number of free fermions is zero. By comparing $F[(\mathrm{Spin}(32)/\mathbb{Z}_2)_1; \langle 8 \rangle]$ with $F[(E_8)_1 \times (E_8)_1; a_1 a_2]$, we find that only their RR components differ by a sign. This means that

$$
F[(\mathrm{Spin}(32)/\mathbb{Z}_2)_1; \langle 8 \rangle] = \overline{(D_8)_1 \times (D_8)_1} \times \mathrm{Arf}. \tag{5.34}
$$

### 5.3.4 By $\mathbb{Z}_2^{\langle 0,16 \rangle_0}$

The discussion again goes in parallel to the previous subsections. The invariant Lie subalgebra is $\mathrm{U}(1) \times A_{15}$. The twisted partition functions are

$$
\begin{aligned}
\mathcal{Z}[(\mathrm{Spin}(32)/\mathbb{Z}_2)_1] &= \sum_{\substack{k=0 \\ k \in 2\mathbb{Z}}}^{6} (\chi^{A_{15}}_{\wedge^k \mathbf{16}} + \chi^{A_{15}}_{\wedge^{k+8}\mathbf{16}})(\chi^{\mathrm{U}(1)}_{k} + \chi^{\mathrm{U}(1)}_{k+8}), \\
\mathcal{Z}[(\mathrm{Spin}(32)/\mathbb{Z}_2)_1]^g &= \sum_{\substack{k=0 \\ k \in 2\mathbb{Z}}}^{6} (-1)^{k/2}(\chi^{A_{15}}_{\wedge^k \mathbf{16}} + \chi^{A_{15}}_{\wedge^{k+8}\mathbf{16}})(\chi^{\mathrm{U}(1)}_{k} + \chi^{\mathrm{U}(1)}_{k+8}), \\
\mathcal{Z}[(\mathrm{Spin}(32)/\mathbb{Z}_2)_1]_g &= \sum_{\substack{k=0 \\ k \in 2\mathbb{Z}}}^{6} (-1)^{k/2}(\chi^{A_{15}}_{\wedge^k \mathbf{16}} + \chi^{A_{15}}_{\wedge^{k+8}\mathbf{16}})(\chi^{\mathrm{U}(1)}_{k+4} + \chi^{\mathrm{U}(1)}_{k+12}), \\
\mathcal{Z}[(\mathrm{Spin}(32)/\mathbb{Z}_2)_1]_g^g &= \sum_{\substack{k=0 \\ k \in 2\mathbb{Z}}}^{6} (-1)^{k/2+1}(\chi^{A_{15}}_{\wedge^k \mathbf{16}} + \chi^{A_{15}}_{\wedge^{k+8}\mathbf{16}})(\chi^{\mathrm{U}(1)}_{k+4} + \chi^{\mathrm{U}(1)}_{k+12}).
\end{aligned}
\tag{5.35}
$$

where $g$ is the generator of $\mathbb{Z}_2^{\langle 0,16 \rangle_0}$. The first two are derived in App. B, and the remaining ones are obtained by further performing modular transformations. Under fermionization,

$$
\begin{aligned}
&\mathcal{Z}[F[(\mathrm{Spin}(32)/\mathbb{Z}_2)_1; \langle 0,16 \rangle_0]]^{\mathrm{NS}}_{\mathrm{NS}} \\
&\quad = (\chi^{A_{15}}_{\wedge^0 \mathbf{16}} + \chi^{A_{15}}_{\wedge^4 \mathbf{16}} + \chi^{A_{15}}_{\wedge^8 \mathbf{16}} + \chi^{A_{15}}_{\wedge^{12}\mathbf{16}})(\chi^{\mathrm{U}(1)}_{0} + \chi^{\mathrm{U}(1)}_{4} + \chi^{\mathrm{U}(1)}_{8} + \chi^{\mathrm{U}(1)}_{12}), \\
&\mathcal{Z}[F[(\mathrm{Spin}(32)/\mathbb{Z}_2)_1; \langle 0,16 \rangle_0]]^{\mathrm{R}}_{\mathrm{NS}} \\
&\quad = (\chi^{A_{15}}_{\wedge^0 \mathbf{16}} - \chi^{A_{15}}_{\wedge^4 \mathbf{16}} + \chi^{A_{15}}_{\wedge^8 \mathbf{16}} - \chi^{A_{15}}_{\wedge^{12}\mathbf{16}})(\chi^{\mathrm{U}(1)}_{0} - \chi^{\mathrm{U}(1)}_{4} + \chi^{\mathrm{U}(1)}_{8} - \chi^{\mathrm{U}(1)}_{12}), \\
&\mathcal{Z}[F[(\mathrm{Spin}(32)/\mathbb{Z}_2)_1; \langle 0,16 \rangle_0]]^{\mathrm{NS}}_{\mathrm{R}} \\
&\quad = (\chi^{A_{15}}_{\wedge^2 \mathbf{16}} + \chi^{A_{15}}_{\wedge^6 \mathbf{16}} + \chi^{A_{15}}_{\wedge^{10}\mathbf{16}} + \chi^{A_{15}}_{\wedge^{14}\mathbf{16}})(\chi^{\mathrm{U}(1)}_{2} + \chi^{\mathrm{U}(1)}_{6} + \chi^{\mathrm{U}(1)}_{10} + \chi^{\mathrm{U}(1)}_{14}), \\
&\mathcal{Z}[F[(\mathrm{Spin}(32)/\mathbb{Z}_2)_1; \langle 0,16 \rangle_0]]^{\mathrm{R}}_{\mathrm{R}} \\
&\quad = (\chi^{A_{15}}_{\wedge^2 \mathbf{16}} - \chi^{A_{15}}_{\wedge^6 \mathbf{16}} + \chi^{A_{15}}_{\wedge^{10}\mathbf{16}} - \chi^{A_{15}}_{\wedge^{14}\mathbf{16}})(\chi^{\mathrm{U}(1)}_{2} - \chi^{\mathrm{U}(1)}_{6} + \chi^{\mathrm{U}(1)}_{10} - \chi^{\mathrm{U}(1)}_{14}).
\end{aligned}
\tag{5.36}
$$

Using the explicit $q$-series expression $\chi^{\mathrm{U}(1)}_{k} = \frac{1}{\eta}\sum_{m \in \mathbb{Z}} q^{(k+Nm)^2/2N}$ where $N = 16$ in our case, the $\mathrm{U}(1)$ components on the right hand side, in terms of $q$-series, are

$$
\begin{aligned}
\chi^{\mathrm{U}(1)}_{0} + \chi^{\mathrm{U}(1)}_{4} + \chi^{\mathrm{U}(1)}_{8} + \chi^{\mathrm{U}(1)}_{12} &= \frac{\theta_3}{\eta}, & \chi^{\mathrm{U}(1)}_{0} - \chi^{\mathrm{U}(1)}_{4} + \chi^{\mathrm{U}(1)}_{8} - \chi^{\mathrm{U}(1)}_{12} &= \frac{\theta_4}{\eta}, \\
\chi^{\mathrm{U}(1)}_{2} + \chi^{\mathrm{U}(1)}_{6} + \chi^{\mathrm{U}(1)}_{10} + \chi^{\mathrm{U}(1)}_{14} &= \frac{\theta_2}{\eta}, & \chi^{\mathrm{U}(1)}_{2} - \chi^{\mathrm{U}(1)}_{6} + \chi^{\mathrm{U}(1)}_{10} - \chi^{\mathrm{U}(1)}_{14} &= 0,
\end{aligned}
\tag{5.37}
$$

which are precisely the twisted partition functions of 2 free Majorana-Weyl fermions.

In summary, we have

$$
F[(\mathrm{Spin}(32)/\mathbb{Z}_2)_1; \langle 0,16 \rangle_0] = 2\psi \times \overline{(A_{15})_1}.
\tag{5.38}
$$

The theory $\overline{(A_{15})_1}$ is invariant under stacking with Arf, as this operation can be compensated for by an outer automorphism of $A_{15} = \mathrm{SU}(16)$.

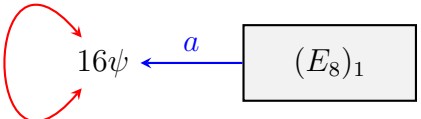

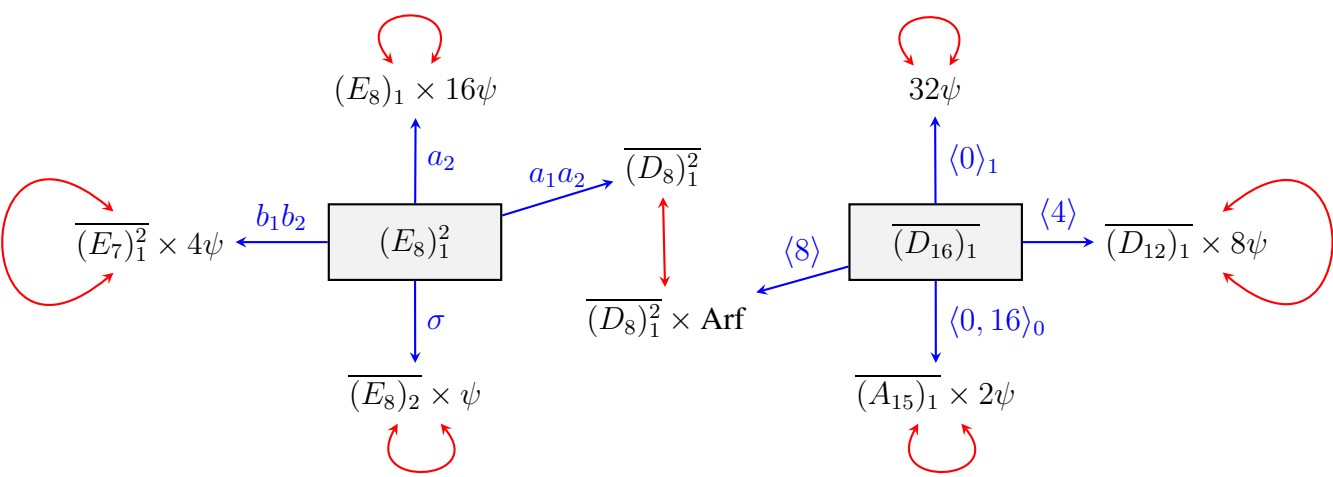

Figure 3: Interrelations among chiral CFTs with $c_L = 8$ and $c_L = 16$. The theories are distinguished by the maximal amount of Majorana-Weyl fermions contained and the maximal affine symmetry contained in the remainder. The blue arrows indicate fermionizations specified by certain $\mathbb{Z}_2$ symmetries, while the red arrows specify the stacking of the Arf theory. Bosonic theories are boxed, and their stacking with Arf is omitted despite being a different theory.

## 5.4 Webs of chiral fermionic CFTs with $c = 8$ and $c = 16$

In Sec. 5.1, we started with a $c = 8$ chiral bosonic CFT $(E_8)_1$ and fermionized it with a non-anomalous $\mathbb{Z}_2$ symmetry to get a chiral fermionic CFT $16\psi$. Further considering how the $c = 8$ theories transform under stacking a fermionic SPT Arf leads to a two node web involving $(E_8)_1$ and $16\psi$ as shown in the upper panel of Fig. 3. We do not consider stacking an Arf on bosonic theories throughout.

In Sec. 5.2 and Sec. 5.3, we started with $c = 16$ chiral bosonic CFTs $(E_8)_1 \times (E_8)_1$ and $(\mathrm{Spin}(32)/\mathbb{Z}_2)_1$ and fermionized them with various anomaly free $\mathbb{Z}_2$ symmetries, to get various chiral fermionic CFTs. We can also consider stacking an Arf invariant on all the $c = 16$ chiral fermionic CFTs, and they form a web as shown in the lower panel of Fig. 3.

All the chiral fermionic CFTs with $c \leq 16$ are included in Fig. 3, from which we can extract the building blocks, as summarized in Table 1. In short, in ascending order of the central charge,

the building blocks are [23]

$$\text{Arf}, \quad \psi, \quad (E_8)_1, \quad \overline{(D_{12})_1}, \quad \overline{(E_7)_1 \times (E_7)_1},$$
$$\overline{(A_{15})_1}, \quad \overline{(E_8)_2}, \quad \overline{(D_8)_1 \times (D_8)_1}, \quad \overline{(D_{16})_1}. \tag{5.39}$$

We also remind the reader that stacking an Arf on the bosonic blocks $(E_8)_1, \overline{(D_{16})_1}$ as well as on $\overline{(D_{12})_1}$ and $\overline{(D_8)_1 \times (D_8)_1}$ gives distinct theories, while for the remaining blocks stacking an Arf can be undone by outer automorphisms. It is then possible to enumerate all the chiral fermionic CFTs. For instance, when $0 < c < 8$, because the only building blocks are Arf and $\psi$, and moreover $\psi$ absorbs Arf, the only possible chiral fermionic CFT is $2c\psi$.

So far, we classified the $c \leq 16$ chiral CFTs and only discussed the webs for $c = 8$ and $c = 16$. How about the webs for the other $c$? Note that when $c \neq 0 \bmod 8$, there is an anomaly in summing over the spin structure, hence it seems that the only topological manipulation is stacking an Arf. In an upcoming work [BSZ24], we will find that a modified bosonization operation can be defined when $c = 4 \bmod 8$ and we also explore the webs for $c \neq 0 \bmod 8$.

# 6 Connections to heterotic strings

We end this paper by discussing the implications of our findings on the classification of heterotic string theories in ten dimensions. We start by recalling the history surrounding their discovery.

**History:** Heterotic string theories were discovered in [GHMR85a] first in the spacetime super-symmetric cases, in the formulation where the $c_L = 16$ part was realized in terms of chiral bosons on even self-dual lattices. An alternative fermionic formulation for the same set of theories was already mentioned in [GHMR85a] whose detail was then provided in [GHMR85b, GHMR86]. These constructions were soon generalized to spacetime non-supersymmetric cases in a short span of time, in e.g. [AGGMV86, DH86, SW86, KLT86]. Table 1 of [KLT86] in particular nicely summarized the known models at that time, to which no other models have been added since then.[24]

The constructions in those early years, however, rest on explicit constructions, either using alternative GSO projections in the fermionic descriptions [AGGMV86, SW86, KLT86], using chiral bosons on odd self-dual lattices [DH86, LNS87], or using explicit higher-level current algebra conformal blocks [FHPV88]. The textbook account in [Pol07] was based on the first approach.

It was therefore not at all clear that all possibilities were exhausted. For example, we can think of starting from 32 Majorana-Weyl fermions, which has $\text{Spin}(32)$ symmetry. We can then pick a complicated non-Abelian finite subgroup and perform an orbifold, possibly with nonzero

---

[23]In accordance with $\overline{(D_{16})_1} = (\text{Spin}(32)/\mathbb{Z}_2)_1$, the fermionic theories $\overline{(D_{12})_1}$, $\overline{(E_7)_1 \times (E_7)_1}$, $\overline{(A_{15})_1}$ and $\overline{(D_8)_1 \times (D_8)_1}$ can also be more explicitly denoted by $H_1$ where the states in the NS-NS sector transform in the genuine representation of $H$, which are $(\text{Spin}(24)/\mathbb{Z}_2)_1$, $((E_7 \times E_7)/\mathbb{Z}_2)_1$, $(\text{SU}(16)/\mathbb{Z}_4)_1$ and $((\text{Spin}(16) \times \text{Spin}(16))/(\mathbb{Z}_2 \times \mathbb{Z}_2))_1$, respectively. See [KOTY23] for a recent discussion.

[24]The T-duality between supersymmetric and non-supersymmetric heterotic strings was also uncovered afterwards, see e.g. [GV87, IT87].

discrete torsion. We can also start from a lattice construction and try to orbifold by a subgroup of the lattice automorphism group. More recently, we learned that there can also be non-invertible symmetries, some of which can be used for orbifolding. Therefore, there was no guarantee that no new models could arise in this manner.[25]

**Generalities on heterotic strings:**  Our analysis in the previous sections can be used to conclude that there are no new exotic models remaining to be found. This can be seen in the following manner.

The perturbative heterotic string theory can be formulated in its most general context [Wit12] by starting from a unitary fermionic CFT of central charge $(c_L, c_R) = (26, 15)$, coupling it to the appropriate ghost and superghost systems, and integrating over the supermoduli space of $\mathcal{N}=(0, 1)$ super Riemann surfaces. Ten-dimensional heterotic string theories form a special case where the worldsheet CFT with $(c_L, c_R) = (26, 15)$ is given by a product of the $\mathcal{N}=(0, 1)$ sigma model on a ten-dimensional manifold $M_{10}$ with $(c_L, c_R) = (10, 15)$ and a fermionic CFT $T$ with $(c_L, c_R) = (16, 0)$.

Integration over the supermoduli space of $\mathcal{N}=(0, 1)$ super Riemann surfaces automatically implements a GSO projection, i.e. the summation over the spin structure associated to the supercharge of the $\mathcal{N}=(0, 1)$ superconformal symmetry. In the fermionic formulation of heterotic string theories, it often happens that many other GSO projections, i.e. summations over the spin structure associated to fermions not affecting that of the supercharge, are performed. From the perspective of the most general construction of heterotic string theories, such additional GSO projections can be thought to be already performed in defining the fermionic CFT $T$ with $(c_L, c_R) = (16, 0)$ on the worldsheet.

Therefore, the classification of ten-dimensional heterotic string theories reduces to the classification of chiral fermionic CFTs with $c_L = 16$. From our results, it is easy to conclude that the list of such CFTs are as given in Table 2. This agrees with the heterotic portion of Table 1 of [KLT86], showing that the list of models found in the olden days is actually complete. The reader may also find it useful to compare Fig. 3 above to [BD97, Figure 1].

---

[25]In this paper, we concentrate on the worldsheet formulations of heterotic string theory. We can also pose the same question from the spacetime point of view, at least when we assume $\mathcal{N}=1$ supersymmetry. In this case, the Green-Schwarz anomaly cancellation, originally found in the foundational paper [GS84], restricts the gauge group to satisfy various conditions, such as the requirement that its dimension should be $496$, and the requirement that a particular quartic Casimir should be the square of a quadratic Casimir. It was known already at the time the textbook [GSW88] was written that the only possible gauge group $G$ is $E_8 \times E_8$, SO(32), $E_8 \times U(1)^{248}$ or $U(1)^{496}$. The details of this rather tedious process can be found e.g. in [Ant15]. It was later found in [ADT10] that the structure of the Chern-Simons modification of the $B$-field dictated by the Green-Schwarz mechanism is compatible with $\mathcal{N}=1$ supersymmetry for $E_8 \times E_8$ or SO(32) but not for $E_8 \times U(1)^{248}$ or $U(1)^{496}$. Therefore, the spacetime analysis tells us that the only anomaly-free gauge group in a ten-dimensional $\mathcal{N}=1$ supergravity system is either $E_8 \times E_8$ or SO(32). It is however not straightforward to generalize this to non-supersymmetric cases.

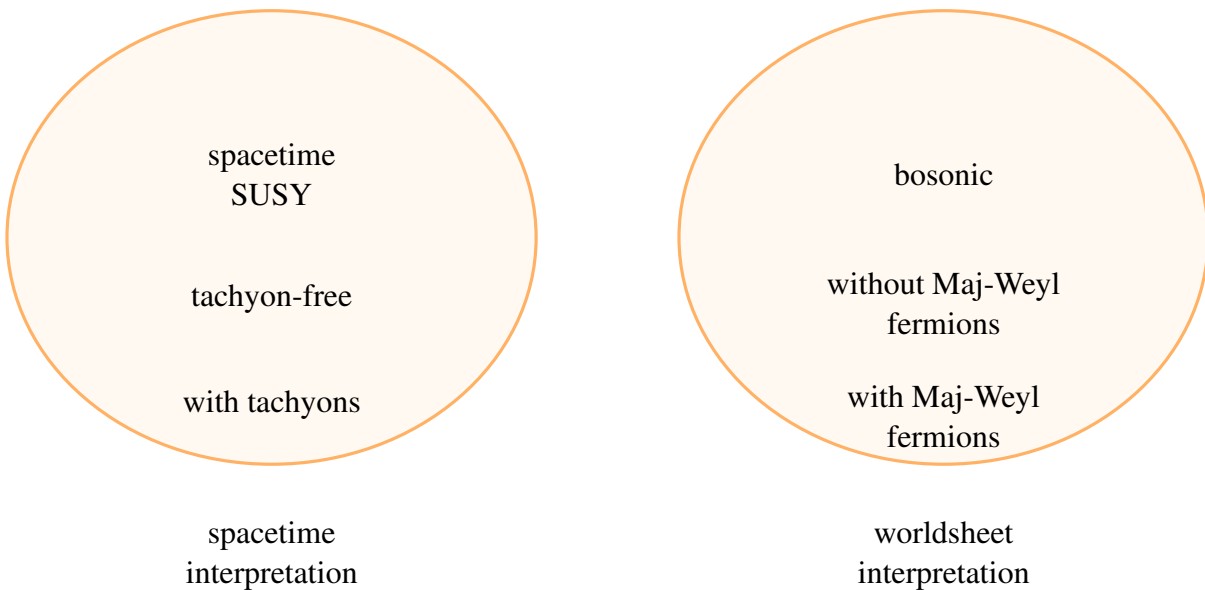

Figure 4: Properties of the spacetime theories v.s. properties of the worldsheet theory $T$. Bosonic worldsheet theories lead to spacetime supersymmetry; worldsheet theories without Majorana-Weyl fermions lead to tachyon-free spacetime theories; and worldsheet theories with Majorana-Weyl fermions lead to spacetime theories with tachyons.

**Properties of the worldsheet theory vs. properties of the spacetime theory:**  The properties of the ten-dimensional spacetime theories can be easily mapped to the properties of the worldsheet fermionic CFT $T$ of $c_L = 16$. The result is summarized in Table 2. The salient points are also summarized in Fig. 4.

We start from the NS sector. The eigenvalues of $L_0$ are integral or half-integral.

- The identity state at $L_0 = 0$ leads to the graviton, the dilaton and the $B$-field.

- The states at $L_0 = 1/2$ give the spacetime tachyons. The OPE of $L_0 = 1/2$ operators in a unitary theory is very constrained and they are forced to be free Majorana-Weyl fermions. Therefore the number of tachyons can be easily read off from the number of Majorana-Weyl fermion factors in the theory $T$.

- The states at $L_0 = 1$ give the massless vector bosons. Indeed, on the worldsheet, the OPE of the operators with $L_0 = 1$ are constrained to satisfy the Jacobi identity, giving the spacetime gauge group.

- Finally, the states at $L_0 \geq 3/2$ give massive stringy modes.

We next discuss the R-sector. The eigenvalues of $L_0$ are integral.

- A state at $L_0 = 0$ gives a spacetime gravitino and an accompanying dilatino, making the theory spacetime supersymmetric. Such a dimension-0 state in the R-sector can be used to

Table 2: List of ten-dimensional heterotic string theories.

| theory | | tachyons | gravitino +dilatino | $\Psi_+$ | $\Psi_-$ |
|---|---|---|---|---|---|
| | $32\psi$ | 32 | | none | none |
| $(E_8)_1$ | $\times 16\psi$ | 16 | | $\mathbf{1} \otimes \mathbf{128}_S$ | $\mathbf{1} \otimes \mathbf{128}_C$ |
| $\overline{(D_{12})_1}$ | $\times\ 8\psi$ | 8 | | $\mathbf{24} \otimes \mathbf{8}_S$ $+C \otimes \mathbf{8}_C$ | $\mathbf{24} \otimes \mathbf{8}_C$ $+C \otimes \mathbf{8}_S$ |
| $\overline{(E_7)_1} \times \overline{(E_7)_1}\ \times\ 4\psi$ | | 4 | | $\mathbf{56} \otimes \mathbf{2}_S$ $+\mathbf{56}' \otimes \mathbf{2}_C$ | $\mathbf{56} \otimes \mathbf{2}_C$ $+\mathbf{56}' \otimes \mathbf{2}_S$ |
| $\overline{(A_{15})_1}$ | $\times\ 2\psi$ | 2 | | $\wedge^2\mathbf{16} \otimes \mathbf{1}_S$ $+\overline{\wedge^2\mathbf{16}} \otimes \mathbf{1}_C$ | $\wedge^2\mathbf{16} \otimes \mathbf{1}_C$ $+\overline{\wedge^2\mathbf{16}} \otimes \mathbf{1}_S$ |
| $\overline{(E_8)_2}$ | $\times\ 1\psi$ | 1 | | $\mathbf{adj}$ | $\mathbf{adj}$ |
| $\overline{(D_8)_1 \times (D_8)_1}$ | | | | $\mathbf{16} \times \mathbf{16}'$ | $S + S'$ |
| $\overline{(D_{16})_1}$ | | | 1 | $\mathbf{adj}$ | |
| $(E_8)_1 \times (E_8)_1$ | | | 1 | $\mathbf{adj} + \mathbf{adj}$ | |

EXPLANATION OF THE TABLE

- Each 10d heterotic string theory is given by a chiral fermionic CFT with $c_L = 16$, which we give as a combination of a seed part and a number $n_f$ of free Majorana-Weyl fermions.

- The spacetime gauge algebra is the direct sum of the affine part and $SO(n_f)$.

- The number of spacetime tachyons is equal to the number of free Majorana-Weyl fermions, which are in the vector representation of $SO(n_f)$ just mentioned.

- The theory is spacetime supersymmetric when the worldsheet theory is bosonic, corresponding to shaded rows.

- $\Psi_+$ and $\Psi_-$ are the gauge representations of the spacetime massless spin-1/2 fermions with positive and negative chirality. We typically use the dimension to specify an irreducible representation, except for $\wedge^2\mathbf{16}$ for the second-rank antisymmetric tensor of $A_{15} = SU(16)$ and for $S$ and $C$ for $D_n = SO(2n)$. The primes are for the second factor in the affine part. When the $SO(n_f)$ symmetry carried by the tachyons is non-Abelian, the representations are given in the form (affine part) $\otimes$ ($SO(n_f)$ part), where the $SO(n_f)$ part is always a spinor of $SO(n_f)$.

establish a one-to-one map between the NS-sector states and the R-sector states by multiplication, meaning that the worldsheet theory $T$ is actually a bosonic theory (possibly multiplied by the Arf theory). Conversely, any such theory has a single state at $L_0 = 0$ in the R-sector. Therefore, the spacetime supersymmetry follows if and only if the worldsheet theory $T$ is actually bosonic. See Sec. 2.2 for some details about dimension-0 operators in fermionic CFTs.

- States at $L_0 = 1$ give massless spin-$1/2$ fermions. The GSO projection associated with the sum over the spin structure of the worldsheet supercharge correlates the spacetime chirality with the $(-1)^F$ eigenvalues of the internal theory. The gauge representation under which they transform is given in Table 2.

- States at $L_0 \geq 2$ give massive stringy modes.

As a final comment on the main part of this paper, let us note that the gauge representations of the spacetime fermions in the system are almost always distinct between the positive chirality spinors and the negative chirality spinors. Even then, the models with tachyons are always spacetime parity invariant, by performing the spacetime parity and the parity of the $\mathrm{O}(n_f)$ acting on $n_f$ tachyons at the same time. It would be interesting to explore these and related issues on the global structure of the gauge and spacetime symmetry of various heterotic string theories.

## Acknowledgments

The authors thank Justin Kaidi for collaboration in the early phase of the project and also for various helpful comments on the draft.

YL is supported by the Simons Collaboration Grant on the Non-Perturbative Bootstrap. PBS, YT and YZ are supported in part by WPI Initiative, MEXT, Japan at Kavli IPMU, the University of Tokyo. YT is also supported by JSPS KAKENHI Grant-in-Aid (Kiban-S), No.16H06335. YL thanks the hospitality of the Kavli Institute of Theoretical Physics and the workshop 'Bootstrapping Quantum Gravity'.

## A    Decomposing the $E_8$ lattice

In this appendix, we present the details of how the $E_8$ lattice sum is decomposed into $D_8$ lattice sums under the $\mathbb{Z}_2^a$ symmetry, and $A_1 \times E_7$ lattice sums under the $\mathbb{Z}_2^b$ symmetry. The $E_8$ lattice is

generated by the simple roots $\alpha_1, ..., \alpha_8$, where

$$
\begin{pmatrix} \alpha_1 \\ \alpha_2 \\ \alpha_3 \\ \alpha_4 \\ \alpha_5 \\ \alpha_6 \\ \alpha_7 \\ \alpha_8 \end{pmatrix} = \begin{pmatrix}
1 & -1 & 0 & 0 & 0 & 0 & 0 & 0 \\
0 & 1 & -1 & 0 & 0 & 0 & 0 & 0 \\
0 & 0 & 1 & -1 & 0 & 0 & 0 & 0 \\
0 & 0 & 0 & 1 & -1 & 0 & 0 & 0 \\
0 & 0 & 0 & 0 & 1 & -1 & 0 & 0 \\
0 & 0 & 0 & 0 & 0 & 1 & -1 & 0 \\
0 & 0 & 0 & 0 & 0 & 0 & 1 & 1 \\
-\frac{1}{2} & -\frac{1}{2} & -\frac{1}{2} & -\frac{1}{2} & -\frac{1}{2} & \frac{1}{2} & \frac{1}{2} & -\frac{1}{2}
\end{pmatrix}. \tag{A.1}
$$

The fundamental weights $\omega_1, ..., \omega_8$, satisfying $(\omega_i, \alpha_j) = \delta_{ij}$, are given by

$$
\begin{pmatrix} \omega_1 \\ \omega_2 \\ \omega_3 \\ \omega_4 \\ \omega_5 \\ \omega_6 \\ \omega_7 \\ \omega_8 \end{pmatrix} = \begin{pmatrix}
\frac{1}{2} & -\frac{1}{2} & -\frac{1}{2} & -\frac{1}{2} & -\frac{1}{2} & -\frac{1}{2} & -\frac{1}{2} & \frac{1}{2} \\
0 & 0 & -1 & -1 & -1 & -1 & -1 & 1 \\
-\frac{1}{2} & -\frac{1}{2} & -\frac{1}{2} & -\frac{3}{2} & -\frac{3}{2} & -\frac{3}{2} & -\frac{3}{2} & \frac{3}{2} \\
-1 & -1 & -1 & -1 & -2 & -2 & -2 & 2 \\
-\frac{3}{2} & -\frac{3}{2} & -\frac{3}{2} & -\frac{3}{2} & -\frac{3}{2} & -\frac{5}{2} & -\frac{5}{2} & \frac{5}{2} \\
-1 & -1 & -1 & -1 & -1 & -1 & -2 & 2 \\
-\frac{1}{2} & -\frac{1}{2} & -\frac{1}{2} & -\frac{1}{2} & -\frac{1}{2} & -\frac{1}{2} & -\frac{1}{2} & \frac{3}{2} \\
-1 & -1 & -1 & -1 & -1 & -1 & -1 & 1
\end{pmatrix}. \tag{A.2}
$$

**Decomposition under $\mathbb{Z}_2^a$:** Under the $\mathbb{Z}_2^a$, which corresponds to $x = \frac{1}{2}\omega_7$ in (4.2), the $E_8$ lattice decomposes into the direct sum of $\mathbb{Z}_2^a$-even and $\mathbb{Z}_2^a$-odd lattices, denoted as $(E_8)_{\mathbb{Z}_2^a\text{-even}}$ and $(E_8)_{\mathbb{Z}_2^a\text{-odd}}$ respectively. In terms of generators,

$$
(E_8)_{\mathbb{Z}_2^a\text{-even}} = \langle \alpha_1, ..., \alpha_6, 2\alpha_7, \alpha_8 \rangle, \qquad (E_8)_{\mathbb{Z}_2^a\text{-odd}} = (E_8)_{\mathbb{Z}_2^a\text{-even}} + \alpha_7. \tag{A.3}
$$

Indeed, any vertex operator labeled by a vector in $(E_8)_{\mathbb{Z}_2^a\text{-even}}$ transforms trivially since $e^{2\pi i(x,\alpha)} = 1$ for any $\alpha \in (E_8)_{\mathbb{Z}_2^a\text{-even}}$, and those labeled by a vector in $(E_8)_{\mathbb{Z}_2^a\text{-odd}}$ gets a minus sign due to $e^{2\pi i(x,\alpha_7)} = e^{\pi i(\omega_7,\alpha_7)} = -1$. It is useful to note that

$$
2\alpha_7 = \omega_1 - \sum_{j \neq 7}(A^{E_8})^{-1}_{j,1}\alpha_j, \tag{A.4}
$$

where $(A^{E_8})_{i,j} = (\alpha_i, \alpha_j)$ is the Cartan matrix of the $E_8$ Lie algebra. Since the matrix elements of $(A^{E_8})^{-1}$ are all integers, the $(E_8)_{\mathbb{Z}_2^a\text{-even}}$ lattice can be rewritten as $\langle \alpha_1, ..., \alpha_6, \omega_1, \alpha_8 \rangle$. Our task is to relate the $(E_8)_{\mathbb{Z}_2^a\text{-even}}$ lattice with the $D_8$ lattice.

The $D_8$ lattice is generated by the simple roots $\beta_1, ..., \beta_8$, where

$$
\begin{pmatrix} \beta_1 \\ \beta_2 \\ \beta_3 \\ \beta_4 \\ \beta_5 \\ \beta_6 \\ \beta_7 \\ \beta_8 \end{pmatrix} = \begin{pmatrix} 1 & -1 & 0 & 0 & 0 & 0 & 0 & 0 \\ 0 & 1 & -1 & 0 & 0 & 0 & 0 & 0 \\ 0 & 0 & 1 & -1 & 0 & 0 & 0 & 0 \\ 0 & 0 & 0 & 1 & -1 & 0 & 0 & 0 \\ 0 & 0 & 0 & 0 & 1 & -1 & 0 & 0 \\ 0 & 0 & 0 & 0 & 0 & 1 & -1 & 0 \\ 0 & 0 & 0 & 0 & 0 & 0 & 1 & -1 \\ 0 & 0 & 0 & 0 & 0 & 0 & 1 & 1 \end{pmatrix}. \tag{A.5}
$$

The fundamental weights $\nu_1, ..., \nu_8$, satisfying $(\nu_i, \beta_j) = \delta_{ij}$, are given by

$$
\begin{pmatrix} \nu_1 \\ \nu_2 \\ \nu_3 \\ \nu_4 \\ \nu_5 \\ \nu_6 \\ \nu_7 \\ \nu_8 \end{pmatrix} = \begin{pmatrix} 1 & 0 & 0 & 0 & 0 & 0 & 0 & 0 \\ 1 & 1 & 0 & 0 & 0 & 0 & 0 & 0 \\ 1 & 1 & 1 & 0 & 0 & 0 & 0 & 0 \\ 1 & 1 & 1 & 1 & 0 & 0 & 0 & 0 \\ 1 & 1 & 1 & 1 & 1 & 0 & 0 & 0 \\ 1 & 1 & 1 & 1 & 1 & 1 & 0 & 0 \\ \frac{1}{2} & \frac{1}{2} & \frac{1}{2} & \frac{1}{2} & \frac{1}{2} & \frac{1}{2} & \frac{1}{2} & -\frac{1}{2} \\ \frac{1}{2} & \frac{1}{2} & \frac{1}{2} & \frac{1}{2} & \frac{1}{2} & \frac{1}{2} & \frac{1}{2} & \frac{1}{2} \end{pmatrix}. \tag{A.6}
$$

It is then straightforward to check that

$$
\begin{aligned}
\alpha_i &= \beta_i, \quad i = 1, 2, ..., 6, \\
\alpha_8 &= -\nu_8 + \beta_6 + \beta_7 + \beta_8, \\
\omega_1 &= -\nu_8 + \sum_{i \neq 7} \beta_i.
\end{aligned} \tag{A.7}
$$

Hence

$$
(E_8)_{\mathbb{Z}_2^a\text{-even}} = \langle \alpha_1, ..., \alpha_6, \omega_1, \alpha_8 \rangle = \langle \beta_1, ..., \beta_6, \beta_7, -\nu_8 + \beta_8 \rangle. \tag{A.8}
$$

The $(E_8)_{\mathbb{Z}_2^a\text{-even}}$ lattice is isomorphic to the $D_8$ lattice, where the first seven generators $\beta_i, i = 1, ..., 7$ are shared by both, and the last generator is $\beta_8$ in $D_8$ which is isomorphic to $-\nu_8 + \beta_8$ in $(E_8)_{\mathbb{Z}_2^a\text{-even}}$. To see the isomorphism, note that the inner products are identical, $(\beta_8, \beta_i) = (-\nu_8 + \beta_8, \beta_i)$ for $i = 1, ..., 7$, and $(\beta_8, \beta_8) = (-\nu_8 + \beta_8, -\nu_8 + \beta_8)$ thanks to the identity $(A^{D_8})^{-1}_{8,8} = 2$. As a consequence, the characters given by their lattice sums are identical:

$$
\begin{aligned}
\sum_{\beta \in (E_8)_{\mathbb{Z}_2^a\text{-even}}} q^{\frac{1}{2}(\beta,\beta)} &= \sum_{\lambda_i \in \mathbb{Z}, i=1,...,8} q^{\frac{1}{2}(\sum_{i=1}^7 \lambda_i \beta_i + \lambda_8(-\nu_8 + \beta_8), \sum_{i=1}^7 \lambda_i \beta_i + \lambda_8(-\nu_8 + \beta_8))} \\
&= \sum_{\lambda_i \in \mathbb{Z}, i=1,...,8} q^{\frac{1}{2}(\sum_{i=1}^8 \lambda_i \beta_i, \sum_{i=1}^8 \lambda_i \beta_i)} = \sum_{\beta \in D_8} q^{\frac{1}{2}(\beta,\beta)} = \chi_{\mathbf{1}}^{D_8}.
\end{aligned} \tag{A.9}
$$

In short, $\mathcal{Z}[(E_8)_1]|_{\mathbb{Z}_2^a\text{-even}} = \chi_{\mathbf{1}}^{D_8}$.

To see the summation of the $\mathbb{Z}_2^a$-odd sublattice of $E_8$, note that $(E_8)_{\mathbb{Z}_2^a\text{-odd}} = (E_8)_{\mathbb{Z}_2^a\text{-even}} + \alpha_7$, while $\alpha_7 = \beta_8$ in our basis. Hence

$$(E_8)_{\mathbb{Z}_2^a\text{-odd}} = \langle \beta_1, ..., \beta_6, \beta_7, -\nu_8 + \beta_8 \rangle + \beta_8 = \langle \beta_1, ..., \beta_6, \beta_7, -\nu_8 + \beta_8 \rangle + \nu_8. \tag{A.10}$$

Note that the $D_8$ character $\chi_S$ is given by the lattice sum over the root lattice $\langle \beta_1, ..., \beta_6, \beta_7, \beta_8 \rangle$ shifted by $\nu_8$, and by the same computation as in (A.9), we find that the $(E_8)_{\mathbb{Z}_2^a\text{-odd}}$ lattice sum precisely gives $\chi_S^{D_8}$. In short, $\mathcal{Z}[(E_8)_1]|_{\mathbb{Z}_2^a\text{-odd}} = \chi_S^{D_8}$.

In summary, we find

$$\begin{aligned} \mathcal{Z}[(E_8)_1] &= \mathcal{Z}[(E_8)_1]|_{\mathbb{Z}_2^a\text{-even}} + \mathcal{Z}[(E_8)_1]|_{\mathbb{Z}_2^a\text{-odd}} = \chi_{\mathbf{1}}^{D_8} + \chi_S^{D_8}, \\ \mathcal{Z}[(E_8)_1]^a &= \mathcal{Z}[(E_8)_1]|_{\mathbb{Z}_2^a\text{-even}} - \mathcal{Z}[(E_8)_1]|_{\mathbb{Z}_2^a\text{-odd}} = \chi_{\mathbf{1}}^{D_8} - \chi_S^{D_8}. \end{aligned} \tag{A.11}$$

which will be used in Sec. 5.1.

**Decomposition under $\mathbb{Z}_2^b$:** Under the $\mathbb{Z}_2^b$, which corresponds to $x = \frac{1}{2}\omega_1$ in (4.2), the $E_8$ lattice decomposes into the direct sum of $\mathbb{Z}_2^b$-even and $\mathbb{Z}_2^b$-odd lattices, denoted again as $(E_8)_{\mathbb{Z}_2^b\text{-even}}$ and $(E_8)_{\mathbb{Z}_2^b\text{-odd}}$, respectively. In terms of generators,

$$(E_8)_{\mathbb{Z}_2^b\text{-even}} = \langle 2\alpha_1, \alpha_2, ..., \alpha_8 \rangle, \qquad (E_8)_{\mathbb{Z}_2^b\text{-odd}} = (E_8)_{\mathbb{Z}_2^b\text{-even}} + \alpha_1. \tag{A.12}$$

Indeed, any vertex operator labeled by a vector in $(E_8)_{\mathbb{Z}_2^b\text{-even}}$ transforms trivially since $e^{2\pi i(x,\alpha)} = 1$ for any $\alpha \in (E_8)_{\mathbb{Z}_2^b\text{-even}}$, and those labeled by a vector in $(E_8)_{\mathbb{Z}_2^b\text{-odd}}$ gets a minus sign due to $e^{2\pi i(x,\alpha_1)} = e^{\pi i(\omega_1,\alpha_1)} = -1$. It is again useful to note that $2\alpha_1 = \omega_1 - \sum_{j\neq 1}(A^{E_8})_{j,1}^{-1}\alpha_j$ so that the $(E_8)_{\mathbb{Z}_2^b\text{-even}}$ can be rewritten as $\langle \omega_1, \alpha_2, ..., \alpha_8 \rangle$. Our task is to relate the $(E_8)_{\text{even}}$ lattice with the $A_1 \times E_7$ lattice.

Since we know that the lattice $(E_8)_{\mathbb{Z}_2^b\text{-even}}$ should be generated by an $E_7$ lattice and an $A_1$ lattice that are mutually orthogonal, it is then natural to take the $E_7$ lattice to be generated by $\langle \alpha_2, ..., \alpha_8 \rangle$, and the $A_1$ lattice generated by $\langle \omega_1 \rangle$. Indeed, the Cartan matrix for $\langle \alpha_2, ..., \alpha_8 \rangle$ is that of $E_7$. Summing over $\langle \alpha_2, ..., \alpha_8 \rangle$ gives $\chi_{\mathbf{1}}^{E_7}$, while summing over $\langle \omega_1 \rangle$ gives $\chi_{\mathbf{1}}^{A_1}$. In short, $\mathcal{Z}[(E_8)_1]|_{\mathbb{Z}_2^b\text{-even}} = \chi_{\mathbf{1}}^{A_1}\chi_{\mathbf{1}}^{E_7}$.

To see the summation of $\mathbb{Z}_2^b$-odd sublattice of $E_8$, note that $(E_8)_{\mathbb{Z}_2^b\text{-odd}} = (E_8)_{\mathbb{Z}_2^b\text{-even}} + \alpha_1$. Hence we should also decompose $\alpha_1$ into the sum of mutually orthogonal weights of $A_1$ and $E_7$. Since $A_1$ is of rank 1, its fundamental weight must be proportional to its root, hence the only possibility is $\frac{\omega_1}{2}$. Indeed, $(\frac{\omega_1}{2}, \omega_1) = 1$. Since the weight of $E_7$ must be orthogonal to that of $A_1$, it must be an expansion of the roots $\langle \alpha_2, ..., \alpha_8 \rangle$. Since the weights are defined up to roots, we find

$$\begin{aligned} (E_8)_{\mathbb{Z}_2^b\text{-odd}} &= \langle \alpha_2, ..., \alpha_8 \rangle + \langle \omega_1 \rangle + \alpha_1 \\ &= \left( \langle \alpha_2, ..., \alpha_8 \rangle + \frac{\alpha_2 + \alpha_4 + \alpha_8}{2} \right) + \left( \langle \omega_1 \rangle + \frac{1}{2}\omega_1 \right). \end{aligned} \tag{A.13}$$

Summing over the $E_7$ lattice with a shift $\frac{\alpha_2+\alpha_4+\alpha_8}{2}$ gives rise to the only non-trivial character $\chi_{\mathbf{56}}^{E_7}$, whose spin is $h = \frac{3}{4}$. Summing over the $A_1$ lattice with a shift $\frac{\omega_1}{2}$ gives rise to the only non-trivial character $\chi_{\mathbf{2}}^{A_1}$, whose spin is $\frac{1}{4}$. In short, $\mathcal{Z}[(E_8)_1]|_{\mathbb{Z}_2^b\text{-odd}} = \chi_{\mathbf{2}}^{A_1}\chi_{\mathbf{56}}^{E_7}$.

In summary, we find

$$\begin{aligned}
\mathcal{Z}[(E_8)_1] &= \mathcal{Z}[(E_8)_1]|_{\mathbb{Z}_2^b\text{-even}} + \mathcal{Z}[(E_8)_1]|_{\mathbb{Z}_2^b\text{-odd}} = \chi_{\mathbf{1}}^{A_1}\chi_{\mathbf{1}}^{E_7} + \chi_{\mathbf{2}}^{A_1}\chi_{\mathbf{56}}^{E_7}, \\
\mathcal{Z}[(E_8)_1]^b &= \mathcal{Z}[(E_8)_1]|_{\mathbb{Z}_2^b\text{-even}} - \mathcal{Z}[(E_8)_1]|_{\mathbb{Z}_2^b\text{-odd}} = \chi_{\mathbf{1}}^{A_1}\chi_{\mathbf{1}}^{E_7} - \chi_{\mathbf{2}}^{A_1}\chi_{\mathbf{56}}^{E_7}.
\end{aligned} \tag{A.14}$$

# B  Decomposing the $D_{16}$ lattice

In this appendix, we present the details of how the $D_{16}$ lattice sum is decomposed under various non-anomalous $\mathbb{Z}_2$ symmetries discussed in Sec. 4. The $D_{16}$ lattice is generated by the simple roots $\alpha_1, ..., \alpha_{16}$, where

$$\begin{pmatrix} \alpha_1 \\ \alpha_2 \\ \alpha_3 \\ \alpha_4 \\ \alpha_5 \\ \alpha_6 \\ \alpha_7 \\ \alpha_8 \\ \alpha_9 \\ \alpha_{10} \\ \alpha_{11} \\ \alpha_{12} \\ \alpha_{13} \\ \alpha_{14} \\ \alpha_{15} \\ \alpha_{16} \end{pmatrix} = \begin{pmatrix} 1 & -1 & 0 & 0 & 0 & 0 & 0 & 0 & 0 & 0 & 0 & 0 & 0 & 0 & 0 & 0 \\ 0 & 1 & -1 & 0 & 0 & 0 & 0 & 0 & 0 & 0 & 0 & 0 & 0 & 0 & 0 & 0 \\ 0 & 0 & 1 & -1 & 0 & 0 & 0 & 0 & 0 & 0 & 0 & 0 & 0 & 0 & 0 & 0 \\ 0 & 0 & 0 & 1 & -1 & 0 & 0 & 0 & 0 & 0 & 0 & 0 & 0 & 0 & 0 & 0 \\ 0 & 0 & 0 & 0 & 1 & -1 & 0 & 0 & 0 & 0 & 0 & 0 & 0 & 0 & 0 & 0 \\ 0 & 0 & 0 & 0 & 0 & 1 & -1 & 0 & 0 & 0 & 0 & 0 & 0 & 0 & 0 & 0 \\ 0 & 0 & 0 & 0 & 0 & 0 & 1 & -1 & 0 & 0 & 0 & 0 & 0 & 0 & 0 & 0 \\ 0 & 0 & 0 & 0 & 0 & 0 & 0 & 1 & -1 & 0 & 0 & 0 & 0 & 0 & 0 & 0 \\ 0 & 0 & 0 & 0 & 0 & 0 & 0 & 0 & 1 & -1 & 0 & 0 & 0 & 0 & 0 & 0 \\ 0 & 0 & 0 & 0 & 0 & 0 & 0 & 0 & 0 & 1 & -1 & 0 & 0 & 0 & 0 & 0 \\ 0 & 0 & 0 & 0 & 0 & 0 & 0 & 0 & 0 & 0 & 1 & -1 & 0 & 0 & 0 & 0 \\ 0 & 0 & 0 & 0 & 0 & 0 & 0 & 0 & 0 & 0 & 0 & 1 & -1 & 0 & 0 & 0 \\ 0 & 0 & 0 & 0 & 0 & 0 & 0 & 0 & 0 & 0 & 0 & 0 & 1 & -1 & 0 & 0 \\ 0 & 0 & 0 & 0 & 0 & 0 & 0 & 0 & 0 & 0 & 0 & 0 & 0 & 1 & -1 & 0 \\ 0 & 0 & 0 & 0 & 0 & 0 & 0 & 0 & 0 & 0 & 0 & 0 & 0 & 0 & 1 & -1 \\ 0 & 0 & 0 & 0 & 0 & 0 & 0 & 0 & 0 & 0 & 0 & 0 & 0 & 0 & 1 & 1 \end{pmatrix}. \tag{B.1}$$

The fundamental weights $\omega_1, ..., \omega_{16}$, satisfying $(\omega_i, \alpha_j) = \delta_{ij}$, are given by

$$\begin{pmatrix} \omega_1 \\ \omega_2 \\ \omega_3 \\ \omega_4 \\ \omega_5 \\ \omega_6 \\ \omega_7 \\ \omega_8 \\ \omega_9 \\ \omega_{10} \\ \omega_{11} \\ \omega_{12} \\ \omega_{13} \\ \omega_{14} \\ \omega_{15} \\ \omega_{16} \end{pmatrix} = \begin{pmatrix} 1 & 0 & 0 & 0 & 0 & 0 & 0 & 0 & 0 & 0 & 0 & 0 & 0 & 0 & 0 & 0 \\ 1 & 1 & 0 & 0 & 0 & 0 & 0 & 0 & 0 & 0 & 0 & 0 & 0 & 0 & 0 & 0 \\ 1 & 1 & 1 & 0 & 0 & 0 & 0 & 0 & 0 & 0 & 0 & 0 & 0 & 0 & 0 & 0 \\ 1 & 1 & 1 & 1 & 0 & 0 & 0 & 0 & 0 & 0 & 0 & 0 & 0 & 0 & 0 & 0 \\ 1 & 1 & 1 & 1 & 1 & 0 & 0 & 0 & 0 & 0 & 0 & 0 & 0 & 0 & 0 & 0 \\ 1 & 1 & 1 & 1 & 1 & 1 & 0 & 0 & 0 & 0 & 0 & 0 & 0 & 0 & 0 & 0 \\ 1 & 1 & 1 & 1 & 1 & 1 & 1 & 0 & 0 & 0 & 0 & 0 & 0 & 0 & 0 & 0 \\ 1 & 1 & 1 & 1 & 1 & 1 & 1 & 1 & 0 & 0 & 0 & 0 & 0 & 0 & 0 & 0 \\ 1 & 1 & 1 & 1 & 1 & 1 & 1 & 1 & 1 & 0 & 0 & 0 & 0 & 0 & 0 & 0 \\ 1 & 1 & 1 & 1 & 1 & 1 & 1 & 1 & 1 & 1 & 0 & 0 & 0 & 0 & 0 & 0 \\ 1 & 1 & 1 & 1 & 1 & 1 & 1 & 1 & 1 & 1 & 1 & 0 & 0 & 0 & 0 & 0 \\ 1 & 1 & 1 & 1 & 1 & 1 & 1 & 1 & 1 & 1 & 1 & 1 & 0 & 0 & 0 & 0 \\ 1 & 1 & 1 & 1 & 1 & 1 & 1 & 1 & 1 & 1 & 1 & 1 & 1 & 0 & 0 & 0 \\ 1 & 1 & 1 & 1 & 1 & 1 & 1 & 1 & 1 & 1 & 1 & 1 & 1 & 1 & 0 & 0 \\ \tfrac{1}{2} & \tfrac{1}{2} & \tfrac{1}{2} & \tfrac{1}{2} & \tfrac{1}{2} & \tfrac{1}{2} & \tfrac{1}{2} & \tfrac{1}{2} & \tfrac{1}{2} & \tfrac{1}{2} & \tfrac{1}{2} & \tfrac{1}{2} & \tfrac{1}{2} & \tfrac{1}{2} & \tfrac{1}{2} & -\tfrac{1}{2} \\ \tfrac{1}{2} & \tfrac{1}{2} & \tfrac{1}{2} & \tfrac{1}{2} & \tfrac{1}{2} & \tfrac{1}{2} & \tfrac{1}{2} & \tfrac{1}{2} & \tfrac{1}{2} & \tfrac{1}{2} & \tfrac{1}{2} & \tfrac{1}{2} & \tfrac{1}{2} & \tfrac{1}{2} & \tfrac{1}{2} & \tfrac{1}{2} \end{pmatrix}. \tag{B.2}$$

The trivial module is generated by the simple roots, $\langle \alpha_1, ..., \alpha_{16} \rangle$, while the adjoint module is generated by the simple roots with a $\omega_{16}$ shift, $\langle \alpha_1, ..., \alpha_{16} \rangle + \omega_{16}$.

**Decomposition under $\mathbb{Z}_2^{\langle 4 \rangle}$:** Under the $\mathbb{Z}_2^{\langle 4 \rangle}$ symmetry, the $D_{16}$ lattice decomposes into the direct sum of $\mathbb{Z}_2^{\langle 4 \rangle}$-even and $\mathbb{Z}_2^{\langle 4 \rangle}$-odd lattices, denoted as $(D_{16})_{\langle 4 \rangle\text{-even}}$ and $(D_{16})_{\langle 4 \rangle\text{-odd}}$, respectively. Using $x = \frac{1}{2}\omega_4$ to compute the $\mathbb{Z}_2^{\langle 4 \rangle}$ charges, we find the decompositions

$$
\begin{aligned}
(D_{16})_{\langle 4 \rangle\text{-even}} &= [\langle \alpha_1, ..., \alpha_3, 2\alpha_4, \alpha_5, ..., \alpha_{16} \rangle] \\
&\quad \cup [\langle \alpha_1, ..., \alpha_3, 2\alpha_4, \alpha_5, ..., \alpha_{16} \rangle + \omega_{16}], \\
(D_{16})_{\langle 4 \rangle\text{-odd}} &= [\langle \alpha_1, ..., \alpha_3, 2\alpha_4, \alpha_5, ..., \alpha_{16} \rangle + \alpha_4] \\
&\quad \cup [\langle \alpha_1, ..., \alpha_3, 2\alpha_4, \alpha_5, ..., \alpha_{16} \rangle + \omega_{16} + \alpha_4].
\end{aligned}
\tag{B.3}
$$

It is useful to note that $2\alpha_4 = \omega_2 - \sum_{j \neq 4}(A^{D_{16}})_{2,j}^{-1}\alpha_j$, where $(A^{D_{16}})_{i,j} = (\alpha_i, \alpha_j)$ is the Cartan matrix of the $D_{16}$ Lie algebra. Since the matrix elements of $(A^{D_{16}})_{2,j}^{-1}$ are all integers for any $j$, the $\langle \alpha_1, ..., \alpha_3, 2\alpha_4, \alpha_5, ..., \alpha_{16} \rangle$ sublattice can be rewritten as $\langle \alpha_1, ..., \alpha_3, \omega_2, \alpha_5, ..., \alpha_{16} \rangle$. Our task is to relate $(D_{16})_{\langle 4 \rangle\text{-even}}$ with the $D_4 \times D_{12}$ lattice.

Since we know that the lattice $(D_{16})_{\langle 4 \rangle\text{-even}}$ should be generated by a $D_4$ lattice and a $D_{12}$ lattice that are mutually orthogonal, it is then natural to take the $D_4$ and $D_{12}$ lattices to be generated by

$$
D_4: \quad \langle \alpha_1, \alpha_2, \alpha_3, \omega_2 \rangle, \qquad D_{12}: \quad \langle \alpha_5, ..., \alpha_{16} \rangle.
\tag{B.4}
$$

Since $\omega_{16}$ can be naturally decomposed into $\omega_4^{D_4} \oplus \omega_{12}^{D_{12}}$ where $\omega_4^{D_4} = (\frac{1}{2}, \frac{1}{2}, \frac{1}{2}, \frac{1}{2})$, whose shift on top of the $D_4$ root lattice generates the spinor module of $D_4$, and similarly for $\omega_{12}^{D_{12}}$, summing over $[\langle \alpha_1, ..., \alpha_3, \omega_2, \alpha_5, ..., \alpha_{16} \rangle]$ yields $\chi_{\mathbf{1}}^{D_4}\chi_{\mathbf{1}}^{D_{12}}$, and summing over $[\langle \alpha_1, ..., \alpha_3, \omega_2, \alpha_5, ..., \alpha_{16} \rangle + \omega_{16}]$ yields $\chi_S^{D_4}\chi_S^{D_{12}}$. Hence summing over the $(D_{16})_{\langle 4 \rangle\text{-even}}$ lattice yields $\mathcal{Z}[(\mathrm{Spin}(32)/\mathbb{Z}_2)_1]|_{\langle 4 \rangle\text{-even}} = \chi_{\mathbf{1}}^{D_4}\chi_{\mathbf{1}}^{D_{12}} + \chi_S^{D_4}\chi_S^{D_{12}}$.

To see the decomposition of the $(D_{16})_{\langle 4 \rangle\text{-odd}}$ lattice, we need to further decompose $\alpha_4$. Note that $\alpha_4 = (0, 0, 0, 1) \oplus (-1, 0, ..., 0)$, which is a direct sum of weights of $D_4$ and $D_{12}$, each of which gives rise to the vector module. This means that $\mathcal{Z}[(\mathrm{Spin}(32)/\mathbb{Z}_2)_1]|_{\langle 4 \rangle\text{-odd}} = \chi_{\mathbf{8}}^{D_4}\chi_{\mathbf{24}}^{D_{12}} + \chi_C^{D_4}\chi_C^{D_{12}}$.

In summary, we find

$$
\begin{aligned}
\mathcal{Z}[(\mathrm{Spin}(32)/\mathbb{Z}_2)_1] &= \chi_{\mathbf{1}}^{D_4}\chi_{\mathbf{1}}^{D_{12}} + \chi_S^{D_4}\chi_S^{D_{12}} + \chi_{\mathbf{8}}^{D_4}\chi_{\mathbf{24}}^{D_{12}} + \chi_C^{D_4}\chi_C^{D_{12}}, \\
\mathcal{Z}[(\mathrm{Spin}(32)/\mathbb{Z}_2)_1]^g &= \chi_{\mathbf{1}}^{D_4}\chi_{\mathbf{1}}^{D_{12}} + \chi_S^{D_4}\chi_S^{D_{12}} - \chi_{\mathbf{8}}^{D_4}\chi_{\mathbf{24}}^{D_{12}} - \chi_C^{D_4}\chi_C^{D_{12}}.
\end{aligned}
\tag{B.5}
$$

where $g$ is the generator of $\mathbb{Z}_2^{\langle 4 \rangle}$.

**Decomposition under $\mathbb{Z}_2^{\langle 8 \rangle}$:** Under $\mathbb{Z}_2^{\langle 8 \rangle}$, the $D_{16}$ lattice decomposes as

$$
\begin{aligned}
(D_{16})_{\langle 8 \rangle\text{-even}} &= [\langle \alpha_1, ..., \alpha_7, \omega_2, \alpha_9, ..., \alpha_{16} \rangle] \\
&\quad \cup [\langle \alpha_1, ..., \alpha_7, \omega_2, \alpha_9, ..., \alpha_{16} \rangle + \omega_{16}], \\
(D_{16})_{\langle 8 \rangle\text{-odd}} &= [\langle \alpha_1, ..., \alpha_7, \omega_2, \alpha_9, ..., \alpha_{16} \rangle + \alpha_8] \\
&\quad \cup [\langle \alpha_1, ..., \alpha_7, \omega_2, \alpha_9, ..., \alpha_{16} \rangle + \omega_{16} + \alpha_8],
\end{aligned}
\tag{B.6}
$$

where we used $2\alpha_8 = \omega_2 - \sum_{j\neq 8}(A^{D_{16}})^{-1}_{2,j}\alpha_j$. Repeating the same discussion as the previous cases, we find that summing over the following sublattices on the left hand side gives rise to the corresponding characters on the right hand side,

$$
\begin{aligned}
[\langle\alpha_1, ..., \alpha_7, \omega_2, \alpha_9, ..., \alpha_{16}\rangle] : &\qquad \chi^{D_8}_{\mathbf{1}}\chi^{D_8}_{\mathbf{1}'}, \\
[\langle\alpha_1, ..., \alpha_7, \omega_2, \alpha_9, ..., \alpha_{16}\rangle + \omega_{16}] : &\qquad \chi^{D_8}_{S}\chi^{D_8}_{S'}, \\
[\langle\alpha_1, ..., \alpha_7, \omega_2, \alpha_9, ..., \alpha_{16}\rangle + \alpha_8] : &\qquad \chi^{D_8}_{\mathbf{16}}\chi^{D_8}_{\mathbf{16}'}, \\
[\langle\alpha_1, ..., \alpha_7, \omega_2, \alpha_9, ..., \alpha_{16}\rangle + \omega_{16} + \alpha_8] : &\qquad \chi^{D_8}_{C}\chi^{D_8}_{C'}.
\end{aligned}
\tag{B.7}
$$

In summary, we find

$$
\begin{aligned}
\mathcal{Z}[(\mathrm{Spin}(32)/\mathbb{Z}_2)_1] &= \chi^{D_8}_{\mathbf{1}}\chi^{D_8}_{\mathbf{1}'} + \chi^{D_8}_{S}\chi^{D_8}_{S'} + \chi^{D_8}_{\mathbf{16}}\chi^{D_8}_{\mathbf{16}'} + \chi^{D_8}_{C}\chi^{D_8}_{C'}, \\
\mathcal{Z}[(\mathrm{Spin}(32)/\mathbb{Z}_2)_1]^g &= \chi^{D_8}_{\mathbf{1}}\chi^{D_8}_{\mathbf{1}'} + \chi^{D_8}_{S}\chi^{D_8}_{S'} - \chi^{D_8}_{\mathbf{16}}\chi^{D_8}_{\mathbf{16}'} - \chi^{D_8}_{C}\chi^{D_8}_{C'},
\end{aligned}
\tag{B.8}
$$

where $g$ is the generator of $\mathbb{Z}_2^{\langle 8\rangle}$.

**Decomposition under $\mathbb{Z}_2^{\langle 0,16\rangle_0}$:** Under the $\mathbb{Z}_2^{\langle 0,16\rangle_0}$ symmetry, the $D_{16}$ lattice decomposes into $\mathbb{Z}_2^{\langle 0,16\rangle_0}$ even and odd sublattices, as

$$
\begin{aligned}
(D_{16})_{\langle 0,16\rangle_0\text{-even}} &= [\langle\alpha_1, ..., \alpha_{15}, 2\alpha_{16}\rangle] \cup [\langle\alpha_1, ..., \alpha_{15}, 2\alpha_{16}\rangle + \omega_{16}], \\
(D_{16})_{\langle 0,16\rangle_0\text{-odd}} &= [\langle\alpha_1, ..., \alpha_{15}, 2\alpha_{16}\rangle + \alpha_{16}] \cup [\langle\alpha_1, ..., \alpha_{15}, 2\alpha_{16}\rangle + \omega_{16} + \alpha_{16}].
\end{aligned}
\tag{B.9}
$$

We would like to decompose the $(D_{16})_{\langle 0,16\rangle_0\text{ even/odd}}$ lattice sums into $A_{15}$ and U(1) lattice sums.

For simplicity, we first consider summing over $\langle\alpha_1, ..., \alpha_{15}, 2\alpha_{16}\rangle$. Note that the simple roots $\langle\alpha_1, ..., \alpha_{15}\rangle$ span the $A_{15}$ lattice, so it is useful to write $2\alpha_{16} = \beta + \sum_{i=1}^{15}x_i\alpha_i$, such that $\beta$ is orthogonal to $\alpha_j, j = 1, ..., 15$. This determines

$$
x_i = \left(-\frac{1}{4}, -\frac{1}{2}, -\frac{3}{4}, -1, -\frac{5}{4}, -\frac{3}{2}, -\frac{7}{4}, -2, -\frac{9}{4}, -\frac{5}{2}, -\frac{11}{4}, -3, -\frac{13}{4}, -\frac{7}{2}, -\frac{7}{4}\right),
\tag{B.10}
$$

and

$$
\beta = \left(\frac{1}{4}, \frac{1}{4}, \frac{1}{4}, \frac{1}{4}, \frac{1}{4}, \frac{1}{4}, \frac{1}{4}, \frac{1}{4}, \frac{1}{4}, \frac{1}{4}, \frac{1}{4}, \frac{1}{4}, \frac{1}{4}, \frac{1}{4}, \frac{1}{4}, \frac{1}{4}\right).
\tag{B.11}
$$

Below, we will use $(\beta, \beta) = 1$ and $(\beta, \omega_{16}) = 2$ repeatedly. We would like to evaluate the lattice sum,

$$
\begin{aligned}
\sum_{\substack{n_1,...,n_{15}\in\mathbb{Z}, \\ n_{16}\in\mathbb{Z}}} & q^{\frac{1}{2}((n_i+x_i n_{16})\alpha_i+n_{16}\beta, (n_j+x_j n_{16})\alpha_j+n_{16}\beta)} \\
&= \sum_{n_1,...,n_{15}\in\mathbb{Z}} q^{\frac{1}{2}((n_i+x_i n_{16})\alpha_i, (n_j+x_j n_{16})\alpha_j)}\sum_{n_{16}\in\mathbb{Z}} q^{\frac{1}{2}n_{16}^2},
\end{aligned}
\tag{B.12}
$$

where the repeated indices $i, j = 1, ..., 15$ are summed over. Since $4x_i \in \mathbb{Z}$, summing over $n_{16} \in 4\mathbb{Z}$ can be simplified by redefining the $n_i$, and the above lattice sum simplifies to

$$\sum_{n_1,...,n_{15}\in\mathbb{Z}} q^{\frac{1}{2}(n_i\alpha_i, n_j\alpha_j)} \sum_{n_{16}\in 4\mathbb{Z}} q^{\frac{1}{2}n_{16}^2} = \sum_{n_1,...,n_{15}\in\mathbb{Z}} q^{\frac{1}{2}(n_i\alpha_i, n_j\alpha_j)} \sum_{m\in\mathbb{Z}} q^{8m^2}. \tag{B.13}$$

By using the definitions of $A_{15}$ and $\mathrm{U}(1)$ characters (see e.g. [CS99, Chapter 4])

$$\chi^{A_{15}}_{\wedge^k \mathbf{16}} = \frac{1}{\eta^{15}} \sum_{n_1,...,n_{15}\in\mathbb{Z}} q^{\frac{1}{2}(n_i\alpha_i + \Gamma_k, n_j\alpha_j + \Gamma_k)}, \quad \chi^{\mathrm{U}(1)}_k = \frac{1}{\eta} \sum_{m\in\mathbb{Z}} q^{\frac{1}{32}(k+16m)^2}, \tag{B.14}$$

where $\Gamma_k$ contains $16 - k$ components of $\frac{k}{16}$, and $k$ components of $-\frac{N-k}{16}$. The first lattice sum in (B.13) yields the character $\chi^{A_{15}}_{\wedge^0 \mathbf{16}}$, while the second lattice sum yields $\chi^{\mathrm{U}(1)}_0$. Likewise, summing over $n_{16} \in 4\mathbb{Z} + 1$ yields $\chi^{A_{15}}_{\wedge^{12}\mathbf{16}}$ and $\chi^{\mathrm{U}(1)}_4$, summing over $n_{16} \in 4\mathbb{Z} + 2$ yields $\chi^{A_{15}}_{\wedge^8\mathbf{16}}$ and $\chi^{\mathrm{U}(1)}_8$, and summing over $n_{16} \in 4\mathbb{Z} + 3$ yields $\chi^{A_{15}}_{\wedge^4\mathbf{16}}$ and $\chi^{\mathrm{U}(1)}_{12}$. Hence (B.12) is equivalent to $\chi^{A_{15}}_{\wedge^0\mathbf{16}}\chi^{\mathrm{U}(1)}_0 + \chi^{A_{15}}_{\wedge^4\mathbf{16}}\chi^{\mathrm{U}(1)}_{12} + \chi^{A_{15}}_{\wedge^8\mathbf{16}}\chi^{\mathrm{U}(1)}_8 + \chi^{A_{15}}_{\wedge^{12}\mathbf{16}}\chi^{\mathrm{U}(1)}_4$. Repeating the same exercise for the other components in (B.9) yields

$$
\begin{aligned}
[\langle\alpha_1, ..., \alpha_{15}, 2\alpha_{16}\rangle]: \quad & \chi^{A_{15}}_{\wedge^0\mathbf{16}}\chi^{\mathrm{U}(1)}_0 + \chi^{A_{15}}_{\wedge^4\mathbf{16}}\chi^{\mathrm{U}(1)}_{12} + \chi^{A_{15}}_{\wedge^8\mathbf{16}}\chi^{\mathrm{U}(1)}_8 + \chi^{A_{15}}_{\wedge^{12}\mathbf{16}}\chi^{\mathrm{U}(1)}_4, \\
[\langle\alpha_1, ..., \alpha_{15}, 2\alpha_{16}\rangle + \omega_{16}]: \quad & \chi^{A_{15}}_{\wedge^0\mathbf{16}}\chi^{\mathrm{U}(1)}_8 + \chi^{A_{15}}_{\wedge^4\mathbf{16}}\chi^{\mathrm{U}(1)}_8 + \chi^{A_{15}}_{\wedge^8\mathbf{16}}\chi^{\mathrm{U}(1)}_0 + \chi^{A_{15}}_{\wedge^{12}\mathbf{16}}\chi^{\mathrm{U}(1)}_{12}, \\
[\langle\alpha_1, ..., \alpha_{15}, 2\alpha_{16}\rangle + \alpha_{16}]: \quad & \chi^{A_{15}}_{\wedge^2\mathbf{16}}\chi^{\mathrm{U}(1)}_{14} + \chi^{A_{15}}_{\wedge^6\mathbf{16}}\chi^{\mathrm{U}(1)}_{10} + \chi^{A_{15}}_{\wedge^{10}\mathbf{16}}\chi^{\mathrm{U}(1)}_6 + \chi^{A_{15}}_{\wedge^{14}\mathbf{16}}\chi^{\mathrm{U}(1)}_2, \\
[\langle\alpha_1, ..., \alpha_{15}, 2\alpha_{16}\rangle + \alpha_{16} + \omega_{16}]: \quad & \chi^{A_{15}}_{\wedge^2\mathbf{16}}\chi^{\mathrm{U}(1)}_2 + \chi^{A_{15}}_{\wedge^6\mathbf{16}}\chi^{\mathrm{U}(1)}_6 + \chi^{A_{15}}_{\wedge^{10}\mathbf{16}}\chi^{\mathrm{U}(1)}_{10} + \chi^{A_{15}}_{\wedge^{14}\mathbf{16}}\chi^{\mathrm{U}(1)}_{14}.
\end{aligned}
\tag{B.15}
$$

In summary, we find

$$
\begin{aligned}
\mathcal{Z}[(\mathrm{Spin}(32)/\mathbb{Z}_2)_1] &= \sum_{\substack{k=0 \\ k\in 2\mathbb{Z}}}^{6} (\chi^{A_{15}}_{\wedge^k\mathbf{16}} + \chi^{A_{15}}_{\wedge^{k+8}\mathbf{16}})(\chi^{\mathrm{U}(1)}_k + \chi^{\mathrm{U}(1)}_{k+8}), \\
\mathcal{Z}[(\mathrm{Spin}(32)/\mathbb{Z}_2)_1]^g &= \sum_{\substack{k=0 \\ k\in 2\mathbb{Z}}}^{6} (-1)^{k/2}(\chi^{A_{15}}_{\wedge^k\mathbf{16}} + \chi^{A_{15}}_{\wedge^{k+8}\mathbf{16}})(\chi^{\mathrm{U}(1)}_k + \chi^{\mathrm{U}(1)}_{k+8}),
\end{aligned}
\tag{B.16}
$$

where $g$ is the generator of $\mathbb{Z}_2^{\langle 0,16\rangle_0}$.

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
