# Peer review of "Classification of chiral fermionic CFTs of central charge $\le 16$"

_SciPost Physics_

## Round 1 · Referee Report · Anonymous (Referee 1) · 2023-6-5

Strengths

1- The paper uses modern self-contained language to describe the classification of chiral fermionic CFTs with central charge relevant for heterotic string theories. 2- Throughout the paper, the authors provide various brief reviews of essential facts that the reader should know for a complete understanding of the classification. To highlight two examples: a) A review of the (non-)concerns about dimension-0 operators in section 2.2 b) A review of "Kac's theorem" in section 4.1. 3- Together, points 1- and 2- work to alleviate any potential concerns about loopholes in their classification. 4- The authors also provide a number of well-organized tables for ease of reference. I am an especially big fan of table 2 and figure 4 together.

Weaknesses

1- The paper essentially states its own main weakness at the start: readers interested in holomorphic VOAs more broadly (not just those relevant for heterotic string theory) may find more value in the references listed at the end of Section 1. 2- Given the authors emphasis on pedagogy and future referability, as well as the importance of "factors of Z2s" in this subject more broadly, I find the use of capital letters to denote Lie algebras and statements like "D_16 = SO(32)" to be very confusing. Similarly, the continued use of the physics terminology (Spin(4k)/Z2) without some comment.

Report

The Journal's acceptance criteria are met. The work provides an excellent reference and successfully accumulates previous literature/references and new ideas in one place. I know I will personally be referring to their paper in future work (see Strengths 2 and 4 esp.). The work requires little to zero grammar/typographical changes. It was a pleasure to read.

Requested changes

1- It would be nice to address Weakness 2 with some revision and/or some footnotes. For example, a very brief comment on the meaning of (Spin(4k)/Z2) would be appreciated. Similarly, a comment on the global symmetries of the bosonic theories versus the fermionic theories, perhaps explaining where (-1)^F sits in the symmetry group of the fermionic theory with some short exact sequence etc. 2- (Minor Typographical). Table 2 and, to a lesser extent, table 1, have bullet point lists which look like they are part of the main text. In particular, the bullet points of table 2 look like they are a continuation of the bullet points of "properties of the worldsheet theory..." list. It would be nice to fix this. 3- (Very Minor Typographical). Perhaps a little more space when presenting the (E_8)_2 characters between (5.18) and (5.19).

---

## Round 1 · Referee Report · Anonymous (Referee 2) · 2023-6-28

Strengths

1- This gives a clear physics motivation for the classification results announced. I think it is readily readable by physicists and explains material that is otherwise rather inaccessible in the mathematics literature.

Weaknesses

1- The main technical weakness of this paper is that is rather loosely written. It is attempting to prove a mathematical result, but I would not trust the results based on the analysis presented here: there are too many places where the arguments are more physical than mathematical. There also appears to be a few places where the mathematics could be improved.

2- There is no particular motivation why one would be interested in these theories which are not, actually, modular invariant, and so I would hesitate to call them "chiral conformal field theories". A single chiral fermion is obviously very interesting, but it is not a consistent conformal field theory on its own in what I understand as the standard requirements of a conformal field theory. Similarly, the rest of the models here are distinguished by the their modular properties, but why invariance (up to a phase) is important is not stated. The application in heterotic string theory is of course one, but apart from that?

3- There is a minor weakness in that the authors were not aware of previous results (from 1995) that covered almost all their classification results. There were two simultaneous papers which did substantially more than these authors, but I don't see that as a particular weakness of this work.

Report

I think that this paper represents a new approach to the previously existing problem of classifying fermionic VOAs with a single representation (called "holomorphic" or "self-dual" elsewhere, and here called "chiral CFTs") and so meets essential criterion 3.

I think it generally meets the general criteria, but I do have some requested changes which address places where it doesn't always meet criteria 3 and 4.

Requested changes

1- please insert the word "unitary" in the abstract e.g. "We classify two-dimensional purely chiral unitary ... "

2- please insert the word "unitary" in the Introduction. I think a footnote is not sufficient for the importance of this requirement.

3- I would argue that the $c_L=c_R<1$ fermionic theories have not actually been classified - lists have been produced which are almost certainly exhaustive, but I am not sure there is actually a water-tight proof that they have been classified.

4- I am not sure the authors say that $(-1)^F$ cannot be defined on the R-sector. In Table 1 (can the caption please appear next to the table), for $c=1/2$, in the fermionic minimal model there are two copies of the $h=1/16$ field, one bosonic and one fermionic, with the same character. Have I misunderstood something?

5- I am afraid that I cannot find the argument outlined in the text in section 2.1 in the reference [FH16] - if it is there, it is in a language that I cannot recognise. I can find the bosonic argument (but not the fermionic) in arxiv:1406.7278 - is it possible this is a mistake? If not, then more explanation is needed for me.

6- After eqn (2.3), shouldn't it be "k is any integer" rather than "k is the smallest integer"?

7- The notation ${\cal Z}_A^B$ in equation (2.4) is not defined. I had no trouble understanding it, but it would be helpful - the corresponding notation in (5.1) is defined, so why not here?

8- I have some queries concerning section 2.2. I do not think anything is wrong, but I find it hard always to follow the arguments and know what the assumptions are.

At the outset, it is said that if $T_F$ is the trivial theory, which I take to mean a theory with a single state $|0\rangle$ which is bosonic and in the NS sector, I do not see how quotienting as in (2.1) can double the number of states.

There appears to be an implicit assumption that all the NS weight zero fields commute - "The $n_\text{NS}$ dimension-0 operators form a commutative Frobenius algebra..". I think this is equivalent to saying that the fields are all bosonic. I am not sure why this has to be true.

Again, there appears to be an implicit assumption that the R sector weight 0 fields commute. If they do not, e.g. if there are two weight 0 fermionic fields with ${a,b}=0$, then you need at least a two-dimensional space for them to act in a non-zero fashion and the "universes" do not decouple.

Finally, there appears also to be an assumption that the weight 0 R fields are fermionic, since that is the case in the ARF theory, and this is needed for the statement that $T_F$ is either bosonic or bosonic$\otimes$Arf.

Generally, I found this section to full of implicit assumptions, which I would rather see explicit, with motivations on physical grounds if they are not mathematical facts.

9- There are quite a few places where there is no referencing to statements which are not obvious, at least I would have liked a reference. Eg. in section 3.2, a reference to why (3.7) is true; in section 3.3, why (3.13) is true; a reference to Kac' theorem in 4.1;

---

## Editorial Decision

resubmitted